**DOI: 10.1038/ncomms13810**　**OPEN**

# Observation of spontaneous spin-splitting in the band structure of an n-type zinc-blende ferromagnetic semiconductor

Le Duc Anh[1,2], Pham Nam Hai[3,4] & Masaaki Tanaka[1,4]

Large spin-splitting in the conduction band and valence band of ferromagnetic semiconductors, predicted by the influential mean-field Zener model and assumed in many spintronic device proposals, has never been observed in the mainstream p-type Mn-doped ferromagnetic semiconductors. Here, using tunnelling spectroscopy in Esaki-diode structures, we report the observation of such a large spontaneous spin-splitting energy (31.7–50 meV) in the conduction band bottom of n-type ferromagnetic semiconductor (In,Fe)As, which is surprising considering the very weak s-d exchange interaction reported in several zinc-blende type semiconductors. The mean-field Zener model also fails to explain consistently the ferromagnetism and the spin-splitting energy of (In,Fe)As, because we found that the Curie temperature values calculated using the observed spin-splitting energies are much lower than the experimental ones by a factor of 400. These results urge the need for a more sophisticated theory of ferromagnetic semiconductors.

[1] Department of Electrical Engineering and Information Systems, The University of Tokyo, 7-3-1 Hongo, Bunkyo-ku, Tokyo 113-8656, Japan. [2] Institute of Engineering Innovation, Graduate School of Engineering, The University of Tokyo, 7-3-1 Hongo, Bunkyo-ku, Tokyo 113-8656, Japan. [3] Department of Physical Electronics, Tokyo Institute of Technology, 2-12-1 Ookayama, Meguro, Tokyo 152-0033, Japan. [4] Center for Spintronics Research Network (CSRN), The University of Tokyo, 7-3-1 Hongo, Bunkyo-ku, Tokyo 113-8656, Japan. Correspondence and requests for materials should be addressed to M.T. (email: masaaki@ee.t.u-tokyo.ac.jp).

A s the design of spintronic devices[1–6] depends heavily on the knowledge of the band structure of the magnetic materials, the band structure of ferromagnetic semiconductors (FMSs) has been the central topic of active debates in the last decade. The most widely used theory of the ferromagnetism in FMSs, 'mean-field Zener model'[7,8], which assumes that the Fermi level lies in the conduction band (CB) or valence band (VB) of the host semiconductors, predicted that the s,p-d exchange interactions would induce spin-splitting in these bands, given by

$$\Delta E = (\alpha \text{ or } \beta)N_0 xS, \tag{1}$$

where $\alpha$ and $\beta$ are the s-d and p-d exchange integrals, respectively, $N_0$ is the cation site concentration, $x$ is the magnetic atom concentration, and $S$ is the spin angular momentum of each magnetic atom. This is the most desirable picture, because we can easily design the material properties and functions by using well-established band engineering of semiconductors[9]. However, recent experimental results in (Ga,Mn)As, which is widely recognized as a 'canonical' FMS, have revealed that the Fermi level lies in the Mn-related impurity band (IB)[10–16], whereas the CB and VB of the host material remain nearly non-magnetic (the 'IB conduction picture')[13]. Small spontaneous spin-splitting $\Delta E$ ($\sim 10$ meV) in the VB has been observed only in p-type Mn-doped II–VI based FMS (Cd,Mn)Te quantum wells (QWs), which is much less-practical owing to the very low Curie temperature ($T_C < 4$ K)[17]. These results casted doubts on the validity of the mean-field Zener model as a universal model to describe and predict the magnetic properties of FMSs. The realisation of FMSs with spin-split host semiconductor's band structure at higher temperature range, which will give a decisive impact on the development of semiconductor-based spintronic devices, also remains elusive.

In this article, we report on the observation of large spontaneous spin-splitting in the CB of an n-type zinc-blende (ZB)-type FMS, (In,Fe)As[18], studied by tunnelling spectroscopy using spin-Esaki-diode devices. The mean-field Zener model predicted no n-type ZB-type FMS because the s-d exchange interaction is generally very weak in these materials[19,20]; however, the electron-induced ferromagnetism has been found and unambiguously confirmed in (In,Fe)As and its $T_C$ can be controlled by changing the electron concentration or manipulating the electron wavefunction in (In,Fe)As QWs[18,21,22]. The Fermi level of n-type (In,Fe)As lies in the CB[21,23], which makes it free from the exotic IB conduction picture observed in (Ga,Mn)As. Furthermore, based on the mean-field Zener model, effective $N_0\alpha$ of (In,Fe)As was estimated to be 2.8 eV $\sim 4.5$ eV (refs 21–23), which implies that $T_C$ of the material would reach room temperature if a high enough electron density ($\sim 10^{20}$ cm$^{-3}$) can be achieved. These effective $N_0\alpha$ values are unexpectedly large, thus raising questions on the conventional understanding of the s-d exchange interaction in FMSs and validity of the mean-field Zener model. Direct observation of the spin-dependent band structure of (In,Fe)As thus would give valuable insights into the ferromagnetism in this material and the physics of FMSs in general, as well as give an important basis for device applications using spin-dependent band engineering.

## Results

**Preparation and I–V characteristics of Esaki-diode devices.** The band structure of (In,Fe)As is studied by tunnelling spectroscopy using spin-Esaki-diode devices of n$^+$-(In,Fe)As/p$^+$-InAs. Figure 1a illustrates the diode devices used in this work, whose layered structure from the surface is 50 nm-thick n$^+$-type (In,Fe)As (with or without Be doping)/5 nm-thick InAs/250 nm-thick p$^+$-type InAs:Be (Be acceptor concentration $N_{Be} = 1 \times 10^{18}$

cm$^{-3}$)/p$^+$ type InAs (001) substrate. Two different samples (A and B) were prepared (see Methods, Supplementary Fig. 1, and Supplementary Note 1) with different Fe concentrations (6% and 8%, respectively) and electron densities (by co-doping Be donors of $5 \times 10^{19}$ cm$^{-3}$ in the (In,Fe)As layer of device B), and consequently different $T_C$ (45 and 65 K, respectively). Although the electron density $n$ of the (In,Fe)As layers in these two samples was not able to be measured by Hall effect measurements due to the parallel conduction in the p$^+$ type InAs substrates, the typical $n$ values for (In,Fe)As samples without and with Be co-doping are in the order of $1 \times 10^{18}$ and $1 \times 10^{19}$ cm$^{-3}$, respectively[18,21,22]. The two samples were then fabricated into mesa diodes with 200 μm in diameter (devices A and B) for I–V measurements. The bias polarity was defined so that in positive bias current flows from the p$^+$-type InAs substrate to the n$^+$-type (In,Fe)As layer.

Figure 1b illustrates the band profiles of these spin-Esaki diodes in different ranges of bias voltage $V$. We assume that (In,Fe)As has a band-gap energy $E_g$ and a spin-split CB (we do not take into account spin-splitting of the VB of (In,Fe)As, because it is away from the Fermi level thus irrelevant to the present study). At $V = 0$, owing to the heavy doping, the Fermi level (denoted by $E_F$) lies at $E_n$ above the CB bottom of n$^+$-(In,Fe)As and at $E_p$ below the VB top of p$^+$-InAs. The dI/dV–V curves of spin-Esaki diodes can be divided into three regions, as illustrated in Fig. 1b. At $V < e^{-1}(E_n + E_p)$ ($e$ is the elementary charge), corresponding to region ① (tunnelling region), electrons tunnel from the CB of the n$^+$-(In,Fe)As to the VB of the p$^+$-InAs. Because the tunnelling conductance dI/dV is proportional to the product of the density of states (DOS) of the electrodes, we can probe the CB structure of the n$^+$-type (In,Fe)As from the dI/dV–V curves in this region. As illustrated in Fig. 1c, at low temperatures ($T < T_C$) or under external magnetic fields, the minority spin and majority spin CBs of (In,Fe)As split, which leads to 'kink' structures in the dI/dV–V curve in this region (pointed by two black arrows in Fig. 1c). Although the dI/dV value also depends on the VB structure of the non-magnetic p$^+$-InAs, the VB of p$^+$-InAs (heavy hole and light hole bands) should show negligible change when applying a magnetic field up to 1 T, or when varying the temperature in the range around $T_C$ of the (In,Fe)As layers of the two devices ($0 \sim 65$ K). Thus, we can elucidate the spin-dependent CB structure of (In,Fe)As by investigating the magnetic field and temperature dependence of the 'kink' structure. When the bias voltage $V$ is equal to $e^{-1}(E_p + E_n)$, the (In,Fe)As CB bottom is lifted to the same energy as the p$^+$-InAs VB top and the current due to the CB-to-VB tunnelling is suppressed, leading to negative differential resistance (NDR). At $e^{-1}(E_n + E_p) < V < e^{-1}E_g$, corresponding to region ② (band-gap region), the tunnelling of electrons into the band-gap is forbidden. However, there is possibly a small current caused by two factors; the thermionic current from electrons and holes thermally hopping and diffusing over the barrier at finite temperatures, and the tunnelling of electrons through the gap states, which usually exist in heavy-doped semiconductors. This additional current may weaken or conceal the NDR characteristics. Finally at larger bias voltages $e^{-1}E_g < V$, corresponding to region ③, the occupied states in the CB (or VB) of n$^+$-(In,Fe)As reach the same energies as the unoccupied states in the CB (or VB) of p$^+$-InAs, then diffusive and drift currents start to flow as in normal pn-junction diodes. Thus we call this region ③ diffusion region.

Figure 1d,e show the dI/dV–V curves measured in devices A and B, respectively, at 3.5 K. In both diode devices, we clearly see all the three regions, as indicated by coloured areas in Fig. 1d,e: The border between the tunnelling region ① and the band-gap region ② is marked by a 'bend' (indicated by blue triangles), observed $\sim 60$ mV in device A and 180 mV in device B. Meanwhile, the starting of the diffusion region ③ is indicated

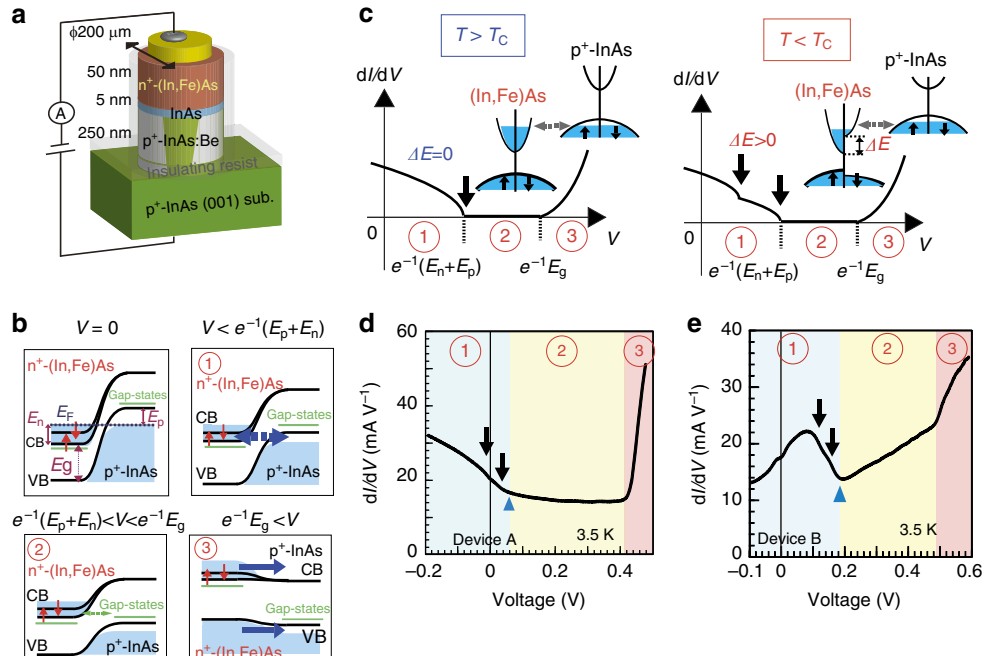

**Figure 1 | Device structure, transport mechanism and d$I$/d$V$–$V$ curves of (In,Fe)As-based Esaki diodes. (a)** Device structure and transport measurement configuration of the spin-Esaki diodes used in this study. (**b**) Band profiles corresponding to the case of zero bias and three regions (①: Tunnelling region, ②: Band-gap region, ③: Diffusion region) of the spin-Esaki diodes. The (In,Fe)As conduction band (CB) is spin-split, and two red arrows represent the up and down spin bands. $E_g$ is the band-gap energy of (In,Fe)As, $E_F$ is the Fermi level, $E_n$, $E_p$ are the quasi-Fermi levels in the $n^+$-(In,Fe)As and $p^+$-InAs electrodes, respectively. Green lines in the band-gap represent gap states coming from heavily-doped impurities. (**c**) Illustration of the change in the band structure of (In,Fe)As and the d$I$/d$V$–$V$ curve of the Esaki diodes at temperature above (left panel) and below (right panel) $T_C$. The spin-splitting in density of states (DOS) of the (In,Fe)As CB leads to the 'kink' structure (pointed by two black arrows) in the d$I$/d$V$–$V$ curve at low temperature ($T < T_C$). (**d,e**) d$I$/d$V$–$V$ characteristics of devices A and B measured at 3.5 K, respectively. Blue, yellow and pink areas correspond to the tunnelling region, band-gap region and diffusion region in each device. Blue triangles indicate the bending positions. Two black arrows point to the kink structures in each d$I$/d$V$–$V$ curve.

by a sharp increase at around 0.42 and 0.48 V in device A and B, respectively.

**Spin-dependent CB structure of (In,Fe)As.** We first focus on the tunnelling regions (region ①) in both devices, where electrons tunnel from the CB of the $n^+$-(In,Fe)As to the VB of the $p^+$-InAs. The kink structures of the d$I$/d$V$–$V$ curves are observed at the end of the tunnelling regions, marked by black arrows in Fig. 1d,e. Clearer splitting structures can be seen at the d$^2I$/d$V^2$–$V$ curves of devices A and B, as shown in Fig. 2a–d. Two-valley splitting structures, corresponding to the kink structures of d$I$/d$V$–$V$ curves, are clearly seen at low temperatures. The two valleys in the d$^2I$/d$V^2$–$V$ curves approach with increasing temperature, and merge above 45 K in device A and 65 K in device B, which agree well with $T_C$ of the (In,Fe)As layers in the two devices. These results strongly support our assignment that these two-valley structures correspond to the spontaneous spin-splitting in the CB of the (In,Fe)As layers.

To analyse these two-valley structures, we use the following fitting function, which is the sum of two Lorentzian curves and a linear offset:

$$\frac{d^2I}{dV^2} = A_{\text{minor}} \frac{\varDelta_{\text{minor}}/2}{(V - V_{\text{minor}})^2 + (\varDelta_{\text{minor}}/2)^2}$$
$$+ A_{\text{major}} \frac{\varDelta_{\text{major}}/2}{(V - V_{\text{major}})^2 + (\varDelta_{\text{major}}/2)^2} + BV + C \quad (2)$$

Here $A_{\text{minor}}$ ($A_{\text{major}}$) is the magnitude, $\varDelta_{\text{minor}}$ ($\varDelta_{\text{major}}$) is the full width at half maximum, $V_{\text{minor}}$ ($V_{\text{major}}$) is the centre bias voltage

of the valley corresponding to the minority (majority) spin CB, respectively, and $BV + C$ is a linear offset. We note that the linear slope $B$ is needed only in the case of device A because of the linear offset in the vicinity of zero bias of the d$^2I$/d$V^2$–$V$ curves. For device B, the $B$ parameter was set to be 0. The valley centre's positions $V_{\text{minor}}$ and $V_{\text{major}}$, which correspond to the bottom edges of the minority and majority spin CBs of (In,Fe)As, are marked by black dots in Fig. 2a,c and white dots in Fig. 2b,d for each curve. From the difference between $V_{\text{minor}}$ and $V_{\text{major}}$, we estimated the spin-split energy $\varDelta E$ of the (In,Fe)As CB (see Supplementary Fig. 2 and Supplementary Note 2) and plotted as a function of temperature $T$ in Fig. 2e. One can see that large $\varDelta E$ persists up to high temperatures close to $T_C$ of both devices. The error bars of $\varDelta E$ are estimated by summing the standard errors of the fitting parameters $V_{\text{major}}$ and $V_{\text{minor}}$, which are smaller than 1 meV in almost all the data points. We also show in Fig. 2e two Brillouin-function fitting curves (two dotted curves) of $\varDelta E$ calculated with the total angular momentum $J = 5/2$ for the $Fe^{3+}$ state, and with $\varDelta E$ (at 3.5 K) and $T_C$ fitted to the experimental values. The red dotted curve was calculated with $T_C = 42$ K and $\varDelta E = 32$ meV (device A, at 3.5 K), and the blue dotted curve was calculated with $T_C = 65$ K and $\varDelta E = 50$ meV (device B, at 3.5 K). Both the Brillouin-function fitting curves explain quite well the temperature dependence of $\varDelta E$ data in devices A and B. However, one can see that the experimental value of $\varDelta E$ does not simply increase proportionally with the Fe concentration $x$: comparing with the $\varDelta E$ and $x$ values of device A, $\varDelta E$ of device B increases by 1.6 times, whereas the increase in $x$ is only 1.3 times (6% for device A and 8% for device B). This

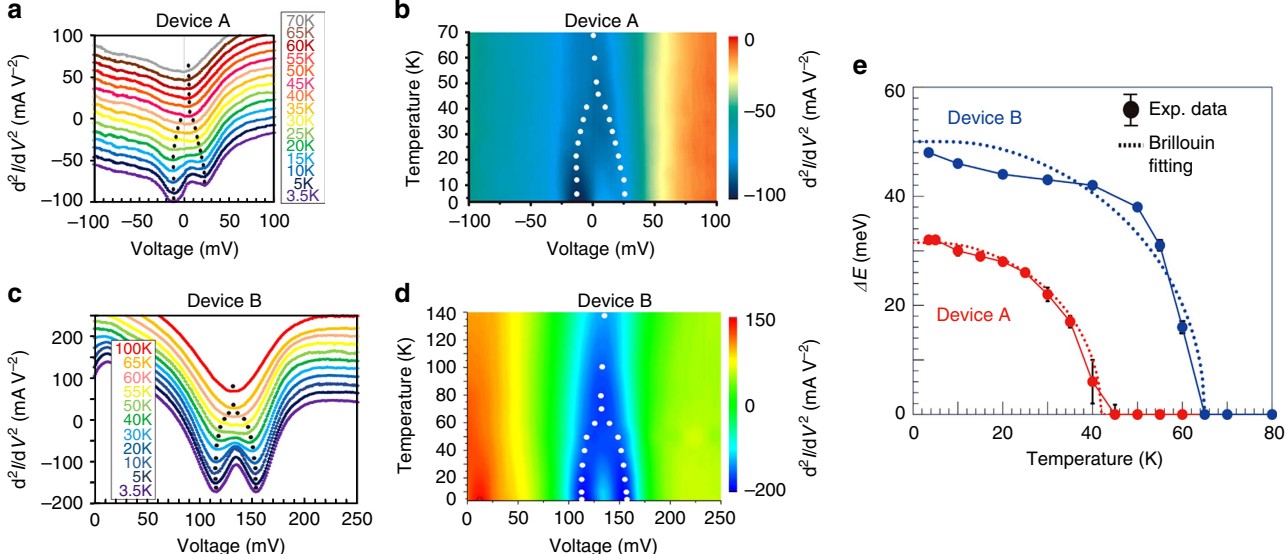

**Figure 2 | Temperature dependence of the spontaneous spin-splitting energy of (In,Fe)As.** (**a** and **c**) $d^2I/dV^2$–$V$ characteristics of devices A and B, respectively, measured at various temperatures (the vertical axes are intentionally shifted for clear vision). (**b** and **d**) Colour plots of the data in **a** and **c**, respectively. White and black dots in **a**–**d** mark the centre positions of the valleys obtained by fitting two Lorentzian curves to the experimental $d^2I/dV^2$–$V$ characteristics. (**e**) Temperature dependence of spin-splitting energy $\Delta E$ in devices A (red circles) and B (blue circles) and their Brillouin-function fitting curves (dotted red and blue curves for devices A and B, respectively).

deviates from the prediction of the mean-field Zener model. A much larger deviation from the mean-field Zener model is the relation between the experimental value of $\Delta E$ and $T_C$ of the same device, which will be discussed later in the Discussion section.

Next, we investigate the dependence of the CB structure of (In,Fe)As on the external magnetic field **H**, applied parallel to the [100] axis in the film plane. The bottom panels of Fig. 3a–d show the $d^2I/dV^2$–$V$ curves (open circles) and their fitting curves (solid curves) of devices A and B measured at various magnetic field strength $H$ ($0 \sim 1$ T), at 3.5 and 50 K, respectively. The top panel of each figure shows the fitting Lorentzian curves, corresponding to the majority spin (solid curves) and minority spin (dotted curves) bands, at 0 and 1 T. The evolution of $\Delta E$ with $H$ in devices A and B are summarized in Fig. 3e,f, respectively (see also Supplementary Figs 3–5 and Supplementary Note 3). At 3.5 K, $\Delta E$ in both devices hardly changes with $H$, indicating a saturated value at low temperature. At 50 K, however, $\Delta E$ increases as $H$ increases in both devices. At 50 K, the (In,Fe)As layer in device A ($T_C = 45$ K) is paramagnetic, whereas that in device B ($T_C = 65$ K) is ferromagnetic. The two spin band structures in device A, which are degenerate at 0 T, start to split and $\Delta E$ reaches 27.7 meV at 1 T. This large $\Delta E$ value corresponds to a giant effective $g$-factor of 478 of (In,Fe)As, which is caused by the strong $s$-$d$ exchange interaction. We note that the measurement temperature of 50 K is close to $T_C$ of device A. In this temperature range, although global ferromagnetic order in device A disappears, local ferromagnetic order possibly still remains that may effectively enhance the $g$-factor. Measurements at temperatures much higher than $T_C$, which are required to accurately estimate the $g$-factor of paramagnetic (In,Fe)As, are difficult because of the broadening of the tunnelling spectra at high temperatures. Meanwhile, $\Delta E$ in device B at 50 K slightly increases from 35 meV at 0 T to 42 meV at 1 T, which is due to the existing ferromagnetic order in the (In,Fe)As layer. It is obvious that the VB of $p^+$-InAs cannot generate this large spin-splitting under a magnetic field of 1 T. Thus, the $\Delta E$–$H$ curves in Fig. 3e,f, together with the $\Delta E$–$T$ curves in Fig. 2e, provide clear evidence that the two-valley structures in the $d^2I/dV^2$–$V$ curves correspond to the majority spin and minority spin CBs of

(In,Fe)As. Note that device A has a 'half-metallic' band structure with 100% of spin polarisation, because the Fermi level lies between the majority and minority spin CB edges.

The temperature range where we can see large spin-split energy in (In,Fe)As CB is limited only by the sample's $T_C$. Therefore, if $T_C$ is increased, these results will open a way to realize FMSs with spin-splitting of the host semiconductor's band structure at high temperature, which are essential for spintronic device applications using spin-dependent band engineering. It is noteworthy that the Fe concentration $x$ (6 and 8%) and electron density $n$ ($\sim 1 \times 10^{19}$ cm$^{-3}$) in the present (In,Fe)As samples are still far below the maximum values achieved in Mn-doped III–V FMSs (the maximum Mn-doping concentration is $\sim 20\%$ (ref. 24) and the maximum hole density is $\sim 10^{21}$ cm$^{-3}$). Thus, there is still much room for increasing either $x$ or $n$, which hopefully leads to higher $T_C$ in (In,Fe)As[18,21–23] as commonly observed in carrier-induced FMSs. The highest $x$ that has been reported so far for (In,Fe)As is 9% (ref. 18), but this can be increased by optimising the growth conditions or using special techniques such as delta doping[25]. On the other hand, the control of $n$ by chemical doping so far has been limited only to the use of Be or Si[18,22]. Although Be or Si atoms are doped in (In,Fe)As, $n$ is limited to at most $\sim 1 \times 10^{19}$ cm$^{-3}$ owing to their amphoteric behaviour and low activation rates in InAs, especially at low growth temperature. Searching for good donors, possibly by using group VI elements or increasing $n$ by electrical gating, are intriguing methods that may increase $n$ to the order of $10^{20}$ cm$^{-3}$.

**Magnetic anisotropy of the band structure of (In,Fe)As.** To properly understand the ferromagnetism of FMSs, we need further information on the band structure of the material, such as the position of the IB, and the magnetic anisotropy. We thus studied tunnelling anisotropic magnetoresistance (TAMR) of (In,Fe)As. Here, the external magnetic field **H** was kept fixed at 1 T and rotated in the film plane, and the $dI/dV$–$V$ curves of device A were measured at various **H** directions with every step of 10 degrees at 3.5 K. At each direction of **H**, we noticed that the $dI/dV$–$V$ curves measured at the magnetic field of 1 T and -1 T

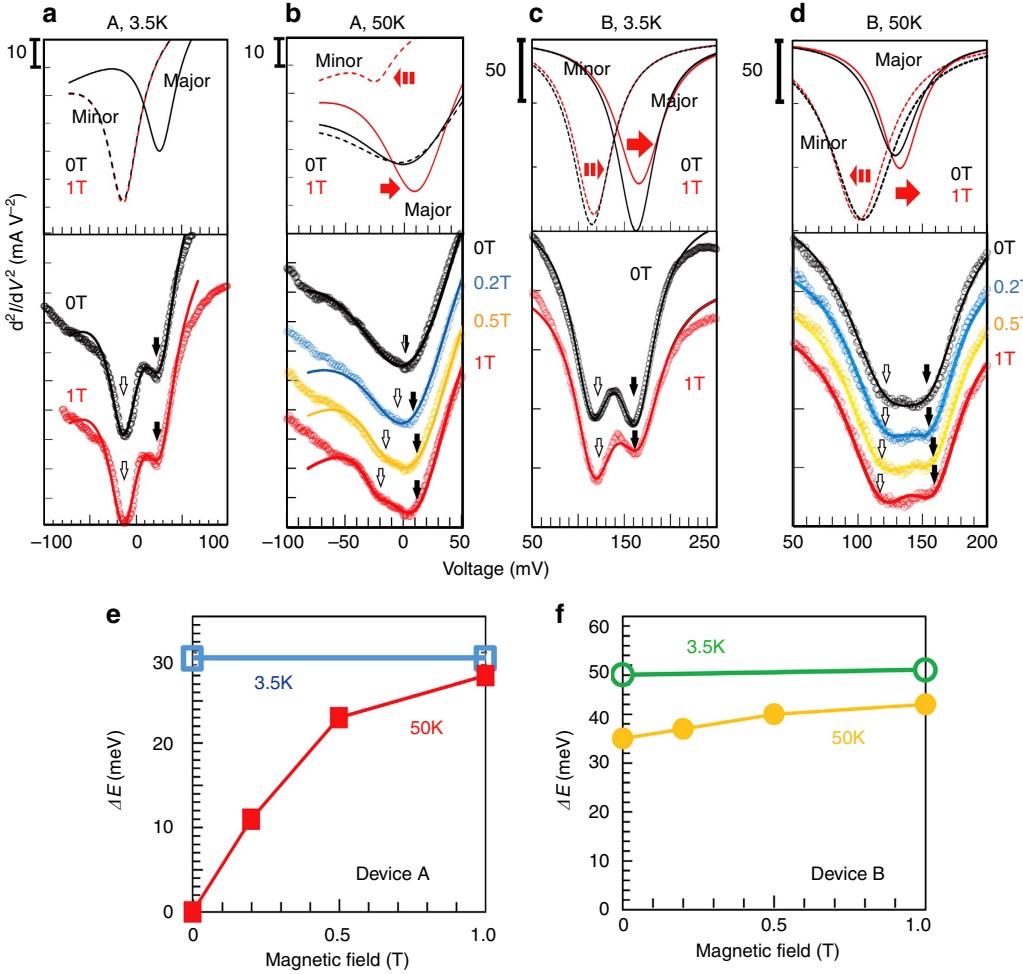

**Figure 3 | Magnetic field dependence of the spin-splitting energy of (In,Fe)As. (a–d)** Evolution of the $d^2I/dV^2$–$V$ curves with external magnetic field strength. Experimental data (open circles) and their fitting curves (solid curves) are shown in the bottom panels, where white and black arrows indicate the minority and majority spin valley centres, respectively (The vertical axes are shifted for clear vision). The fitting curves are the sum of two Lorentzian curves, fitted for each valley. The fitting Lorentzian curves for the data at 0T (black) and 1T (red) are shown in the top panels. **a** and **b** are the results of device A at 3.5 and 50 K, **c** and **d** are the results of device B at 3.5 and 50 K, respectively. (**e** and **f**) Zeeman splitting energies $\Delta E$ at 3.5 and 50 K of devices A (**e**) and B (**f**). The external magnetic field was applied along the [100] axis in the film plane.

are slightly different, which is possibly owing to the contribution of the Hall effect of the current flowing in the p$^+$-InAs substrate (see Supplementary Fig. 6 and Supplementary Note 4). As the magnetoresistance is expected to be an even function of magnetic field, we obtained the d$I$/d$V$–$V$ curve at each direction of **H** by averaging the two d$I$/d$V$–$V$ curves measured at the magnetic field of 1 T and −1 T.

Figure 4a plots the change of $\frac{\mathrm{d}I}{\mathrm{d}V}$

$$\Delta\left(\frac{\mathrm{d}I}{\mathrm{d}V}\right) = \left(\frac{\mathrm{d}I}{\mathrm{d}V} - \left\langle\frac{\mathrm{d}I}{\mathrm{d}V}\right\rangle_\phi\right)\bigg/\left\langle\frac{\mathrm{d}I}{\mathrm{d}V}\right\rangle_\phi \times 100(\%), \quad (3)$$

where $\phi$ is the magnetic field angle from the [100] axis in the counter-clockwise direction in the film plane, and $\left\langle\frac{\mathrm{d}I}{\mathrm{d}V}\right\rangle_\phi$ is the $\frac{\mathrm{d}I}{\mathrm{d}V}$ averaged over $\phi$ at each fixed $V$. We focus on the tunnelling region ① (top panel) and the diffusion region ③ (bottom panel). Note that the colour scales in the two panels are different by two orders of magnitude. No significant change was observed in the band-gap region ② (not shown). One can see the TAMR data in the tunnelling region show a fourfold symmetry and another higher-order (eightfold) symmetry, whereas those in the diffusion region are dominated by twofold terms. The symmetry axis of the twofold symmetry in the diffusion region rotates by 45 degrees

(from [010] to [$\bar{1}$10]) as the bias voltage $V$ increases from 0.42 to 0.49 V. This indicates that there are at least two twofold terms with different symmetry axes in this region.

In Fig. 4b we show the cross-sectional data (blue circles) and the fitting curves (red curves) at six bias points in Fig. 4a: Point 1 ($V = 50\,\mathrm{mV}$) belongs to the tunnelling region, and points 2–6 ($V = 440$–$480\,\mathrm{mV}$) belong to the diffusion region. In the tunnelling region, $\Delta(\frac{\mathrm{d}I}{\mathrm{d}V})$ was fitted by

$$\Delta\left(\frac{\mathrm{d}I}{\mathrm{d}V}\right) = C_{4[100]}\cos(4\phi) + C_{8[100]}\cos(8\phi), \quad (4)$$

where $C_{4[100]}$ and $C_{8[100]}$ are the anisotropy constants of fourfold and eightfold symmetry along the [100] axis, respectively. On the other hand, $\Delta(\frac{\mathrm{d}I}{\mathrm{d}V})$ in the diffusion region can be well fitted by

$$\Delta\left(\frac{\mathrm{d}I}{\mathrm{d}V}\right) = C_{4[100]}\cos(4\phi) + C_{2[010]}\cos\left[2\left(\phi + \frac{\pi}{2}\right)\right] + C_{2[\bar{1}10]}\cos\left[2\left(\phi + \frac{3\pi}{4}\right)\right], \quad (5)$$

where $C_{2[010]}$ and $C_{2[\bar{1}10]}$ are the anisotropy constants of two twofold terms with [010] and [$\bar{1}$10] axes, respectively. The anisotropy constants estimated in the diffusion region are summarized in Fig. 4c.

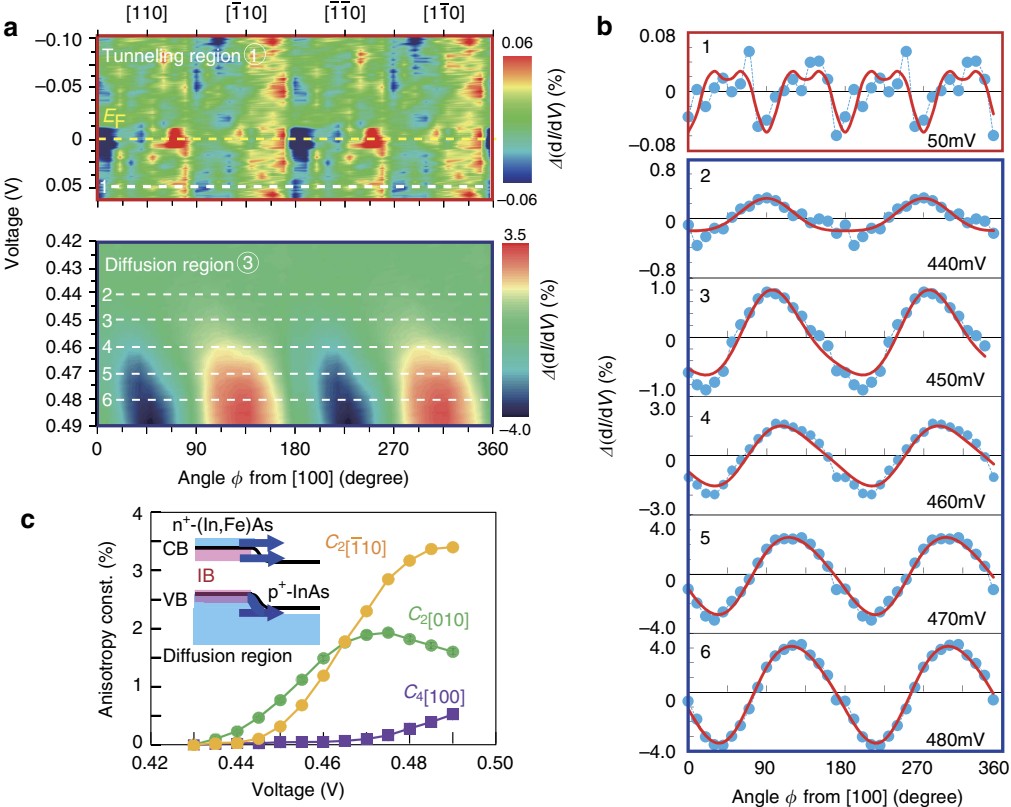

**Figure 4 | In-plane magnetic anisotropy of the (In,Fe)As band structure. (a)** Colour plots of the change of $\frac{dI}{dV}$: $\Delta(\frac{dI}{dV}) = (\frac{dI}{dV} - \langle\frac{dI}{dV}\rangle_\phi)/\langle\frac{dI}{dV}\rangle_\phi \times 100(\%)$ ,

where $\phi$ is the magnetic field angle from the [100] axis in the counter-clockwise direction in the film plane, and $\langle\frac{dI}{dV}\rangle_\phi$ is the $\frac{dI}{dV}$ averaged over $\phi$ at each

fixed $V$, in the tunnelling region (top panel) and the diffusion region (bottom panel) of device A. **(b)** Cross-sectional data of $\Delta(\frac{dI}{dV})$ (blue circles) and fitting

curves (red curves) at six bias points in **a**: Point 1 ($V = 50$ mV) belongs to the tunnelling region, whereas points 2–6 ($V = 440$–480 mV) belong to the

diffusion region. **(c)** Bias voltage dependence of anisotropy constants of fourfold symmetry ($C_{4[100]}$), twofold symmetry along the [010] direction ($C_{2[010]}$),

and twofold symmetry along the [$\bar{1}$10] direction ($C_{2[\bar{1}10]}$), estimated in the diffusion region. The electron flows in the diffusion region are illustrated as blue

arrows in the inset. Possible positions of the Fe-related impurity bands (IBs) relative to the conduction band (CB) and valence band (VB) are also sketched

as pink areas. All the data were measured at 3.5 K.

TAMR originates from the change of the DOS of (In,Fe)As with changing the magnetisation direction, which is caused by the spin-orbit interactions (SOI) and the *s,p-d* exchange interactions[26–28]. Although in the tunnelling region electrons flow only from the (In,Fe)As CB, in the diffusion region electrons flow from the CB, VB and possibly occupied Fe-related IBs of (In,Fe)As to the empty states in the CB and VB of p$^+$-InAs, as illustrated in the inset of Fig. 4c. The different symmetries in the two regions thus come from different components of the (In,Fe)As band structure. The fourfold symmetry likely originates from the CB and VB of (In,Fe)As, which have the cubic symmetry of the zinc-blende crystal structure. The intensity $C_{4[100]}$ in the tunnelling region (0.035%) is much smaller than that in the diffusion region (∼0.3%) because SOI in the CB is much weaker than in the VB. The existence of eightfold symmetry is very unique to (In,Fe)As, and has been observed in the crystalline anisotropic magnetoresistance (AMR) of (In,Fe)As thin films[29]. On the other hand, the origin of the two twofold terms ($C_{2[010]}$ and $C_{2[\bar{1}10]}$) are likely related to the Fe-related IBs. The bias voltages where the fourfold term ($C_{4[100]}$) and the twofold terms ($C_{2[010]}$ and $C_{2[\bar{1}10]}$) appear in the diffusion region are very close, as shown in Fig. 4c, indicating the overlap of these Fe-related IBs with the CB bottom and/or the VB top of (In,Fe)As. This result is consistent with the observed position of the Fe deep levels in InAs, where Fe was doped at a very low concentration[30]. This energy overlap has been proposed to induce the large *s,p-d*

exchange interactions in (In,Fe)As[18,21]. These twofold terms are not observed in the tunnelling region, possibly because the tunnelling is forbidden by the different orbital symmetry of the Fe-related IBs and the p$^+$-InAs VB. This indicates that the Fe-related IB is irrelevant to the spin-splitting observed in the tunnelling region of the two devices A and B.

**Discussion**

The observation of the spontaneous spin-splitting in the CB of (In,Fe)As provides clear evidence of the interaction between electron carriers in the CB and Fe local spins. As both $T_C$ and the spin-splitting energy $\Delta E$ in CB are obtained experimentally, (In,Fe)As can serve as a litmus test for the validity of the mean-field Zener model[7,8]. Using $\Delta E = 31.7$–50 meV (observed), $N_0\alpha$ is estimated to be 0.21 ∼ 0.25 eV in both devices. On the other hand, $T_C$ can be calculated by the following equations:

$$T_C = \frac{S(S+1)}{12k_B N_0} A_F (N_0\alpha)^2 x \rho_{3D} \tag{6}$$

$$\rho_{3D} = \frac{\sqrt{2}m^{*\frac{3}{2}}}{\pi^2 \hbar^3}\sqrt{E_F} \tag{7}$$

Here, $k_B$ is the Boltzmann constant, $A_F$ is the Fermi liquid constant, $x$ is the Fe concentration, $\rho_{3D}$ is the DOS at the Fermi level $E_F$, $\hbar$ is the reduced Planck constant, $m^*$ is the electron

effective mass. Using $m^\star = 0.03 \sim 0.08\, m_0$ ($m_0$ is the free electron mass)[23], $A_F = 1.2$, and $N_0\alpha = 0.21 \sim 0.25$ eV, $T_C$ is calculated to be $0.093 \sim 1.55$ K, which is much lower than the observed $T_C$ of 45–65 K by a factor of $\sim 400$. Inversely, if we estimate the $N_0\alpha$ values from the experimental $T_C$ values of devices A and B and use them to calculate $\Delta E$, it should be above 1 eV, which is much larger than the observed $\Delta E = 31.7$–50 meV. This large discrepancy indicates that the mean-field Zener model is not applicable even to (In,Fe)As, a FMS that is free from the IB conduction picture.

Besides the case of (In,Fe)As, the validity of the mean-field Zener model in explaining the magnetic properties of other FMSs has also been discussed theoretically and experimentally[31–33]. The mean-field Zener model proposed a tendency of higher $T_C$ in wider-gap FMSs such as nitrides and oxides, and it concluded that the ferromagnetism in narrow-gap FMSs should be very weak[7,8]. In fact, however, although the realisation of high-$T_C$ ferromagnetism in (Ga,Mn)N is still challenging, several experimental results have reported strong ferromagnetism in FMSs with narrow-gap hosts such as InAs, GaSb and InSb. For instance, remarkably high $T_C$ ($>300$ K) has been recently realized in molecular beam epitaxy (MBE) grown Fe-doped FMS, (Ga,Fe)Sb[34–36]. Mn-doped narrow-gap FMSs grown by metal organic chemical vapour deposition (MOVPE) such as (In,Mn)As[37,38], (In,Mn)Sb[39] are other intriguing cases that show very high $T_C$. Although the mechanism of these high-$T_C$ ferromagnetism is still not clearly understood, the resonance in energy of the magnetic impurity levels and the CB or VB of the host materials has been proposed as an important factor[21,23,31,33–36], which was not considered in the mean-field Zener model. Building an appropriate unified model for FMSs thus remains an unsolved theoretical challenge.

## Methods

**Sample preparation and characterisations.** All the samples were grown by MBE on p$^+$-type InAs (001) substrates. The p$^+$-type InAs substrates were deoxidized in our MBE growth chamber at 480 °C. Then, a 250 nm-thick Be doped p-type InAs buffer layer and a 5 nm-thick undoped InAs buffer layer were successively grown at 460 °C. The doping level of Be acceptors in the p$^+$-InAs buffer is $1 \times 10^{18}$ cm$^{-3}$. The undoped InAs buffer layer serves as a thin barrier against the diffusion of Be atoms. After cooling the substrate temperature to 236 °C, a 50-nm-thick (In$_{1-x}$Fe$_x$)As thin film was grown. In sample B, we doped Be in the (In,Fe)As layer at a doping level of $5 \times 10^{19}$ cm$^{-3}$. Although Be is a well-known acceptor when replacing the group-III site in III–V semiconductors (including InAs), we have shown that when grown at low temperature (236 °C) by MBE, Be dopants mainly become double-donors in (In,Fe)As layers, probably because they favourably sit in the interstitial sites[18]. We observed bright and streaky *in situ* RHEED patterns during the MBE growth of the (In,Fe)As layers in both samples A and B, indicating good growth conditions and high crystal quality of these samples.

The magnetic properties of the (In,Fe)As layers in the two diode devices were characterized by magnetic circular dichroism and superconducting quantum interference device magnetometry. $T_C$ was estimated to be 45 K in A and 65 K in B, respectively. Details of the sample characterisations are given in Supplementary Note 1.

**Device fabrications and I-V measurements.** We fabricated circular-shaped mesa diodes with 200 µm in diameter using standard photolithography and Ar ion milling. A passivation layer was formed on the sample surface by spin-coating an insulating negative resist (OMR-85, ©Tokyo Ohka). A contact hole of 180 µm in diameter was opened on the top of each mesa by photolithography, and a 200 nm-thick Au electrode with 700 µm in diameter was formed on each mesa by vacuum evaporation and chemical etching. Au wires are bonded to the Au electrode and on the backside of the p$^+$-InAs substrate by In bonding contact for two-terminal transport measurements. The bias polarity was defined so that in the positive bias current flows from the p$^+$-type InAs substrate to the n$^+$-type (In,Fe)As layer (forward bias). d$I$/d$V$–$V$ and d$^2I$/d$V^2$–$V$ characteristics were numerically obtained from the $I$–$V$ data.

**Data availability.** The data that support the findings of this study are available from the authors upon reasonable request.

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

## Acknowledgements

This work is supported by Grants-in-Aid for Scientific Research including the Specially Promoted Research, the Project for Developing Innovation Systems of MEXT, Spin-tronics Research Network of Japan (Spin-RNJ), and the Cooperative Research Project Program of RIEC, Tohoku University. L.D.A. acknowledges the JSPS Fellowship for Young Scientists (No. 257388) and the MERIT Program. P.N.H. acknowledges Kato Foundation for Promotion of Science, Iketani Science and Technology Foundation, and Toray Science Foundation.

## Author contributions

Device fabrication, measurements and data analysis: L.D.A; writing and project planning: L.D.A., P.N.H. and M.T.

## Additional information

**Competing financial interests**: The authors declare no competing financial interests.

**How to cite this article**: Anh, L. D. *et al.* Observation of spontaneous spin-splitting in the band structure of an n-type zinc-blende ferromagnetic semiconductor. *Nat. Commun.* **7,** 13810 doi: 10.1038/ncomms13810 (2016).

**Publisher's note**: 

