## [Peer Review File · Nature Communications]

Reviewers' comments:

Reviewer #1 (Remarks to the Author):

The authors investigate tunnelling spectroscopy in ferromagnetic semiconductor, (In, Fe)As, based Esaki-diode structures at low temperature. From two distinct valley structures in the tunneling region, and the temperature and external magnetic field dependence of these valley structures, they ascribe the structures to the spin splitting in the host conduction band of Fe-doped InAs. This is an interesting result, being different from previous well-known ferromagnetic semiconductors, such as (Ga, Mn)As., and is promising for the engineering offer of magnetic semiconductors for spintronics. However, I would like the authors to address the following points.

1. There are a number of typographical errors e.g.

line 215: "calls for reconsiderations on the chemical trend" should be "calls for reconsideration of the chemical trend"

line 229: "donors in (In,Fe)As layer", should probably be "donors in the (In,Fe)As layer",

I would suggest a careful proof-reading of the manuscript and the supplementary material. On another note, the features in highlighted by the black arrows Figure 1, c and d, are quite subtle, "kink" might be a more fitting description than "step-like".

2. A number of the equations presented in the work seem to be somewhat incomplete or haphazard.

e.g. The in-line equation in line 33, $\Delta E = (\alpha \text{ or } \beta)N_0xS$ the variable x , related to the spin concentration, is not declared.

More alarmingly the equation used in the discussion, line 206 $T_C = S(S+1)/(12k_B N_0) A_F [(N_0 \alpha)]^2 x \rho_{3D}$, again, x appears without being declared, however \hbar and m^* are declared without being mentioned. It can be assumed m^* and \hbar are related to ρ once the relevant reference have been read, but these need to be made explicit. Such errors do not inspire confidence for such crucial calculations.

3. In Fig. 2e, the temperature dependence of ΔE , shows roughly ΔE of 48 meV and 40 meV for devices B and A, respectively, with corresponding TC of 65 K and 45 K. This is actually consistent with the prediction of mean-field Zener model: $T_C \propto (N_0 \alpha)^2 \propto (\Delta E)^2$ (if the differences in x are ignored). A more careful comparison between the observation of ΔE Vs T and theoretical prediction from mean-field Zener model would be more convincing.

4. In Fig. 3c, the majority valley becomes smaller under larger magnetic field; in Fig. a, c and d, the majority valleys are smaller than minority valley, which is not explained. The plots making the comparison between the different field strengths is interesting, could the data from -1T to +1T be made available in the supplementary material.

5. In Fig. 3e the magnetic field dependence of ΔE for device A is in a paramagnetic state at 50 K, why is the linear behavior of the Zeeman Effect is not observed?

6. The section dealing with the analysis of the "Magnetic anisotropy of the band structure of (In,Fe)As" need a bit more care. Could the difference in the curves of +1T and -1T fields be due to sample alignments? The reason for fitting the experimental is rather poorly introduced and seems to be more justified after consideration of facts.

Curiously Fig 4(b) 1 which shows the cross sectional from Fig.4(a) data at 50mV, in the tunnelling region of sample A, the raw data does not appear to be as symmetric as the curve fit. Is there a plausible explanation for the asymmetric behaviour of the peaks?

Reviewer #2 (Remarks to the Author):

I very much enjoyed reading the manuscript by Anh, Hai, and Tanaka. This work follows up on this groups pioneering work to measure the exchange splitting of the electronic band structure of III-V transition metal doped materials using tunneling spectroscopy. In the case of InFeAs, because the

material is an n-type ferromagnet, then the authors are able to construct an interband tunnel diode. Under forward bias (Easki mode), they can tunnel from the CB states in the ferromagnet into the VB states in the p-InAs. By measuring the dI/dV of this tunneling signal, they observe the spin splitting of the CB states, which tracks the magnetization. The measured splitting is too small to account for the T_c using existing theory of magnetism in DMSs (Zener model). The data are very convincing and of high quality. The central conclusions (failure of Zener model to account for the observed ferromagnetism and CB spin splitting) are convincing.

I am supportive of the manuscript for eventual publication in Nature Communications given its high impact in the field of spintronics and magnetic semiconductors. However, the current version contains many technical concerns, enumerated below. I apologize for the length of the report. I would have been able to make this more concise given additional time so please forgive any unintentional redundancies. The comments are roughly in order of appearance in the manuscript. Where appropriate, page number and line numbers are given.

- 1) Is the InFeAs n^{++} or n-type? I think this distinction is important.
- 2) Abstract: authors mention that weak nature of s-d exchange is a "common belief". I have to ask them to reword this statement, because s-d exchange is well measured in II-VI and III-V semiconductors over the last few decades, so it is not a belief, but a result of many measurements.
- 3) Page 2, line 39, the authors cited Kobayashi et al for the proof of the Fermi level position in the impurity band of Mn. I recommend in addition they cite Chapler et al PRB 87, 205314 (2013) because that work shows the clear presence of impurity band and Fermi level in GaMnAs.
- 4) Page 2, line 44, the authors mention that the lack of observation of valence band spin splitting casts doubt on the validity of the mean-field Zener model. I think this is too strongly worded because the mean-field Zener model does accurately predict the ferromagnetism in semiconductors, however the measurements indicate that it does not apply to some materials, so its range applicability is narrower than previously thought.
- 5) Page 3, line 74, can the authors also please include the electron density (n) for each sample?
- 6) Page 4, lines 83-86, authors explain tunneling region, however they don't explain why the tunneling has a spin dependence (important for general Nature audience).
- 7) Page 4, lines 88-89, the authors mention that there should be no current in the band gap region, however, they are not mentioning the possibility of thermionic current from electrons and holes thermally hopping and diffusing over the barrier. Their explanation is only valid at 0K.
- 8) Page 4, line 92, authors state that diffusion region occurs once the applied bias is larger than the built-in bias, however again there is some mistake in this explanation. At finite temperature, electrons and holes undergo diffusive forward bias current due to thermal diffusion over the barrier. At 0K, diffusion current only occurs at $V_{\text{builtin}} = V_{\text{applied}}$, i.e. flat band. Once $V_{\text{applied}} > V_{\text{builtin}}$, then the current is no longer diffusive, but changes to majority drift.
- 9) Page 5, lines 106-109, authors are explaining how the I-V behavior can probe the spin-split DOS, however there is not much explanation here. For general audience of Nature, it is important to explain this concept. Please provide a cartoon to explain how dI/dV probes DOS and spin-split DOS.
- 10) Page 5, lines 117-118, the authors conclude that the two-valley structure corresponds to the spin-splitting of the CB. This is the central premise of the study, and I fully agree with this conclusion.
- 11) Page 5, lines 121-123, the authors discuss that the two valleys are fitted to extract the spin splitting Δ_E , however they don't provide the fit function. They should at a minimum provide the fit function, explain its physical justification, how many fit parameters are used, and what is the uncertainty of the extracted parameters, i.e. error bars. This is especially important for the data near T_c , where clearly the broadening of the peaks means that the fit uncertainty must necessarily diverge.

12) Page 6, lines 124-125, the authors state that the spin splitting of device B is larger than A because of the higher Fe concentration. However, they don't mention what theory or model they base that prediction on. The authors say that mean-field Zener doesn't apply, yet they still want to say that the T_c scales with Fe doping. They should explain what theory supports their statement.

13) Page 6, lines 135-141, the authors explain the magnetic field dependence of the spin splitting device A at 50K, and they claim that the data indicate a g-factor of 621, which is a giant g-factor induced in InFeAs which is larger than observed in II-VI DMSs. Here I actually strongly disagree with the authors. Their M vs T data are too difficult to fit the T_c accurately, therefore they don't really know if all ferromagnetism in sample A is gone by 50K. In fact, there is plenty of spontaneous magnetization at 50K in their data. Thus the anomalously large g-factor is likely just because 50K isn't high enough to make that sample exhibit pure paramagnetism. Only if the authors did the field dependence at higher temperature (where there is no M_{spont}) and fitted with Brillouin function could this be believable. I am not convinced that InFeAs shows anomalously large s-d exchange based on the data shown.

14) Page 6, lines 145-147, authors state that there is still room to increase T_c in InFeAs, but this rather vague. The authors should be more specific and say how much more room there is to increase Fe doping or increase the n-type doping. Specifically, what are the defect limits? Solubility limits?

15) Page 8, lines 183-187, the authors conclude that the 4-fold symmetry in the tunneling region is much smaller than observed in the diffusion region because SOI is weaker than in the VB. But this conclusion is contradictory to the authors' main conclusion, which is that s-d exchange is larger than p-d exchange, however here they are stating the opposite. To be more specific in this section, the authors are stating that the VB spin splitting is larger than the CB splitting, which necessarily requires that p-d exchange is greater than s-d. However, their main conclusion is that p-d exchange = 0, and s-d is anomalously large. This contradiction needs to be resolved.

16) Page 9, lines 201-202, the authors state that s-d exchange is generally considered to be very weak. The wording should be changed, the s-d exchange is well measured in a wide range of DMSs, so it is not considered to be any value, it is just well measured. Rather than using such unqualified comparisons, the authors should directly state, we measure XXX s-d exchange which is different than the range of s-d exchange XX splitting observed in all other DMSs.

17) Page 9, lines 204-205, the authors should note that the $N0_{\alpha}$ values they observed are basically identical to the values observed in all II-VI DMSs. This means that the s-d exchange they measure is not surprisingly large, but actually exactly what would be expected based on measurements in II-VI DMSs carried out a few decades ago.

18) Page 9, lines 213-214, the authors state that the measured s-d exchange splitting is not large enough to account for the observed T_c based on the mean-field Zener model. I agree with this statement. However, I would suggest that rather than stating that the Zener model is a failure (when in fact it explains magnetism in a number of DMSs), they can state that InFeAs in fact have a different physical mechanism for the magnetism, which the Zener model does not capture.

19) Page 10, lines 230-232, the authors are explaining how Be doping is used as both a donor and acceptor. This explanation needs to be a bit more detailed because of the general Nature audience. They need to explain that Be is amphoteric and can sit on either In or As site depending on the substrate temperature and therefore lead to either n or p type doping.

20) Fig2a-d, why are the vertical axes arbitrary units? In Fig. 1 the authors have dI/dV in units of mA/V , therefore d^2I/dV^2 should be units of mA/V^2 .

21) Fig2e, the ΔE values need to have error bars to reflect the uncertainty in the fitting procedure.

22) Fig. 3a-d, the vertical axis now has units of A/V^2 , however the axis is not labeled, so we can't actually know what size of the signal. They should include axes labels. Also, the plotted fitting curves don't appear to be offset Lorentzians, but more complex functions. It is appropriate for the authors to include the fit function in the manuscript.

23) Fig. 4c, if authors carried out fitting, the data points should have error bars to show the

uncertainty of the fit parameters.

Reviewer #3 (Remarks to the Author):

There has been a long standing issue of what is the mechanism for stabilization of ferromagnetism in ferromagnetic semiconductors (FMS). Dietl and Ohno proposed that a mean field (MF) Zener model explains the ferromagnetic behavior of FMS (Ref. 1 and Ref. 2). An equation modeling the behavior was proposed that described the magnetic properties in GaMnAs quite well. However predictions for other FMS were less than satisfactory. Curie temperatures of GaMnN, InMnAs and InMnSb differed from experimental observation as much as several hundred degrees. While Dietl and Ohno continue to stand by the theory, the experimental community does not. Ahn et al investigate the magnetotransport characteristics of InFeAs Esaki diodes using tunneling spectroscopy. From spectroscopic analysis spin-splitting energies of 40-50 meV were measured. This is quite large and the authors indicate that this is the reason why large T_c is observed in InFeAs. The observed T_c can't be explained by the MF Zener model. Thus the authors conclude a different model is needed but do not offer an alternative. The work is of interest to the FMS and wider magnetics communities.

The reviewer concludes that either the theory is wrong or the measurements are wrong or possibly both. The use of tunneling spectroscopy data is fraught with experimental issues. First the band structures of both the conduction band and valence band are needed. The authors assume that the valence and conduction bands are parabolic. They ignore the contributions to the density of states from the heavy hole band and light hole band in the valence band. Note k.p calculations of the valence band of InAs and InMnAs have been published see M. A. Meeker et al PRB 2015. Some discussion is needed regarding tunneling and band structure. Error analysis of tunneling spectroscopy data is needed. Also iron in InAs may have several charge states. The authors should discuss this since it may modify the tunneling spectra.

As to alternative theories, Huang and Wessels noted that Fe in InAs is resonant with the conduction band see reference K. Huang J. Appl. Phys 64 6770 1988. From this they concluded that a vacuum referred binding energy (VRBE) model is relevant for transition metal doping of InAs and other III-V semiconductors. A model was subsequently proposed that transition metals with d-levels resonant with the semiconductor conduction or valence band should be a good FMS with high Curie temperatures (B. Wessels, New Journal of Physics 2008). Semiconductors with transition metals with d levels well within the band gap will not be good FMS as in the case of GaMnN. The InFeAs alloy studied here seems to support this VRBE model since the authors claim that Fe level is resonant with the conduction band.

The breakdown of the Zener model was discussed by Wessels, New Journal of Physics 2008. It is somewhat puzzling that the authors ignore the large body of literature on the InMnAs system in their introduction which has shown high T_c behavior. Also there is prior literature of Fe levels in InAs that should be discussed in light of their work.

The main conclusion is that there is major disagreement between the MF Zener theory of Dietl and Ohno and tunneling spectroscopy results presented in this work.

Other comments:

There is always confusion with possible magnetic precipitates in the Fe-As system. Are there any?

Note Be is an acceptor in III-V semiconductors see typo on line 73.

Responses to the Reviewers' Comments

Le Duc Anh^{1,2}, Pham Nam Hai^{3,4}, and Masaaki Tanaka^{1,4}

¹*Department of Electrical Engineering and Information Systems, The University of Tokyo, 7-3-1 Hongo, Bunkyo-ku, Tokyo 113-8656, Japan*

²*Institute of Engineering Innovation, Graduate School of Engineering, The University of Tokyo, 7-3-1 Hongo, Bunkyo-ku, Tokyo 113-8656, Japan*

³*Department of Physical Electronics, Tokyo Institute of Technology, 2-12-1 Ookayama, Meguro, Tokyo 152-0033, Japan*

⁴*Center for Spintronics Research Network (CSRN), The University of Tokyo, 7-3-1 Hongo, Bunkyo-ku, Tokyo 113-8656, Japan*

First of all, we would like to thank all the reviewers for their valuable and constructive comments, which helped us improve the quality of our paper. In the following, we address and answer all the comments and questions, point by point. We also show revised parts in the revised main manuscript and Supplementary Information. (In the revised main manuscript, the revised parts are colored.)

Re-evaluation of the spin split energy ΔE of device A

The two spin-Esaki diode devices (A and B) studied in this work differ in Fe concentration (6% and 8%, respectively) and electron density (due to co-doping of Be double donors in the (In,Fe)As layer in device B). Both of the two diode devices show two-valley structures in the $d^2I/dV^2 - V$ curves (Figs. 2a-d and Figs. 3a-d in the main manuscript), corresponding to the splitting of the majority spin conduction band (CB) and minority spin CB of (In,Fe)As. For the two-valley structures, we fitted the sum of two Lorentzian curves to determine the valley center positions of the majority and minority spin CBs (V_{major} and V_{minor} , respectively) [see eq.(R5) in page R20]. The spin split energy ΔE of the (In,Fe)As layers was estimated by the difference between V_{major} and V_{minor} . We found that this estimation is appropriate in device B, but needs to be corrected in device A as explained below. Note that, however, the main conclusions remain unchanged.

Figures R1a and b show the band profiles of the p-n junctions in the two devices A and B, respectively, at low temperature (ferromagnetic state) and bias voltage $V = 0$, V_{major} , and V_{minor} . In these figures, we set the Fermi energy at the zero point, and denote the energy levels of the majority and minority spin CB bottom edges of (In,Fe)As and the valence band (VB) top of p^+ -InAs as E_{major} , E_{minor} , and E_p , respectively. The spin split energy ΔE of (In,Fe)As is therefore given by $E_{\text{minor}} - E_{\text{major}}$.

Fig R1 (=Supplementary Figure S2). Band profiles of the p-n junctions in the two devices A (a) and B (b), respectively, at bias voltage $V = 0$, V_{major} , and V_{minor} , all at low temperature (ferromagnetic state). At $V = 0$, with the Fermi energy at the zero point, the energy levels of the majority and minority spin CB bottom edges of (In,Fe)As and of the VB top of p^+ -InAs are denoted as E_{major} , E_{minor} , and E_p , respectively. The blue arrows show the tunnelling directions of electrons at non-zero bias V . In device A, the tunnelling directions of electrons at $V = V_{\text{major}}$ and at $V = V_{\text{minor}}$ are opposite to each other.

In device A, because V_{major} is positive whereas V_{minor} is negative (data shown in Figs. 2a,b of the main manuscript), the Fermi level E_F lies above the band edge of the majority spin CB and below that of the minority spin CB (“half-metallic” band structure, i.e. $E_{\text{major}} < 0$, $E_{\text{minor}} > 0$), as illustrated in Fig. R1a. At V_{major} , the band edge of majority spin CB of (In,Fe)As is aligned with the top of the VB of p^+ -InAs, thus $eV_{\text{major}} = -E_{\text{major}} + E_p$. However, at V_{minor} , which is negative, the band edge of the minority spin CB of (In,Fe)As is aligned with the quasi-Fermi level of the p^+ -InAs, thus $eV_{\text{minor}} = -E_{\text{minor}}$. Therefore, we have the following relation:

$$e(V_{\text{major}} - V_{\text{minor}}) = E_{\text{minor}} - E_{\text{major}} + E_p = \Delta E + E_p \quad (\text{R1})$$

This means that the difference between V_{major} and V_{minor} overestimates the spin split energy ΔE of (In,Fe)As CB in device A by E_p .

To obtain the accurate value of E_p , we calibrated the Be flux of our MBE system by growing one control sample composed of, from the surface, 500 nm-thick GaAs:Be/50

nm-thick GaAs grown on semi-insulating GaAs (001) substrate. The hole density of the GaAs:Be in the control sample was measured to be $1.02 \times 10^{18} \text{ cm}^{-3}$, thus in our $\text{p}^+\text{-InAs}$ with the same Be concentration, $E_p = 8.3 \text{ meV}$ [we used the effective masses of heavy hole and light hole to be $0.41m_0$ and $0.026m_0$, respectively, reported in W. Nakwaski *et al. Physica B* **210**, 1-25 (1995)]. The new ΔE values of device A were obtained by subtracting 8.3 meV from the $e(V_{\text{major}} - V_{\text{minor}})$ values of device A and are shown in Fig. R2 (=Revised Fig. 2e in the revised manuscript). Note that this correction does not affect any main conclusions of our paper.

On the other hand, in device B, because both V_{major} and V_{minor} are positive (data shown in Figs. 2c,d of the main manuscript), the Fermi level E_F in (In,Fe)As lies above both the majority and minority spin CB bottom edges as illustrated in Fig. R1b (thus, $E_{\text{major}} < 0$ and $E_{\text{minor}} < 0$). At V_{major} (V_{minor}), the band edge of the majority (minority) spin CB of (In,Fe)As is aligned with the top of the VB of $\text{p}^+\text{-InAs}$. Therefore we have the following relations:

$$eV_{\text{major}} = -E_{\text{major}} + E_p \quad (\text{R2})$$

$$eV_{\text{minor}} = -E_{\text{minor}} + E_p \quad (\text{R3})$$

$$e(V_{\text{major}} - V_{\text{minor}}) = E_{\text{minor}} - E_{\text{major}} = \Delta E \quad (\text{R4})$$

Thus the difference between V_{major} and V_{minor} corresponds exactly to the spin split energy ΔE of (In,Fe)As CB in device B. No correction for device B is needed.

Fig. R2 (= Revised Fig. 2e in the revised manuscript) ΔE data of devices A and B as a function of temperature, in which the ΔE data of device A was corrected by subtracting E_p (8.3 meV) from the observed $e(V_{\text{major}} - V_{\text{minor}})$ values.

Corresponding revised parts:

- ✧ We corrected the ΔE data of device A in revised Figs. 2e and 3e in the revised manuscript. The effective g -factor of (In,Fe)As estimated at 50 K is therefore corrected to 478 (page 8, line 178).
- ✧ We added Fig. R1 as Supplementary Figure 2 and the estimation of ΔE from V_{major} and V_{minor} as Supplementary Note 2 to Supplementary Information.

Responses to reviewer #1:

1) There are a number of typographical errors e.g.

line 215: "calls for reconsiderations on the chemical trend" should be "calls for reconsideration of the chemical trend"

line 229: "donors in (In,Fe)As layer", should probably be "donors in the (In,Fe)As layer", I would suggest a careful proof-reading of the manuscript and the supplementary material. On another note, the features in highlighted by the black arrows Figure 1, c and d, are quite subtle, "kink" might be a more fitting description than "step-like".

Our response:

Corresponding revised parts:

- ✧ We corrected the typographical errors as indicated in line 215 and 229. The words "step-like" were replaced by "kink" in line 100 (page 4), line 107 (page 5), 129, 133 (page 6) of the revised manuscript.

2. A number of the equations presented in the work seem to be somewhat incomplete or haphazard. E.g. The in-line equation in line 33, $\Delta E = (\alpha \text{ or } \beta)N_0xS$, the variable x , related to the spin concentration, is not declared. More alarmingly the equation used in the

discussion, line 206: $T_C = \frac{S(S+1)}{12k_B N_0} A_F (N_0 \alpha)^2 x \rho_{3D}$, again, x appears without being

declared, however \hbar and m^* are declared without being mentioned. It can be assumed m^* and \hbar are related to ρ_{3D} once the relevant reference have been read, but these need to be made explicit. Such errors do not inspire confidence for such crucial calculations.

Our response:

Corresponding revised parts:

We declared the variable x as the concentration of the magnetic atoms in the in-line equation in page 2, line 38 of the revised manuscript.

- ✧ In page 12, line 275 of the revised manuscript, we added the equation (2) next to

the equation (1) as follows:

$$T_C = \frac{S(S+1)}{12k_B N_0} A_F (N_0 \alpha)^2 x \rho_{3D} \quad (1)$$

$$\rho_{3D} = \frac{\sqrt{2} m^{*2}}{\pi^2 \hbar^3} \sqrt{E_F} \quad (2)$$

Here, S is the spin angular momentum of each magnetic atom, k_B is the Boltzmann constant, N_0 is the cation density, A_F is the Fermi liquid constant, x is the Fe concentration, ρ_{3D} is the density of states (DOS) at the Fermi level E_F , \hbar is the reduced Planck constant, m^* is the electron effective mass. In the revised manuscript, these are explicitly described.

3) In Fig. 2e, the temperature dependence of ΔE , shows roughly ΔE of 48 meV and 40 meV for devices B and A, respectively, with corresponding T_C of 65 K and 45 K. This is actually consistent with the prediction of mean-field Zener model: $T_C \sim (N_0 \alpha)^2 \sim \Delta E^2$ (if the differences in x are ignored). A more careful comparison between the observation of ΔE vs T and theoretical prediction from mean-field Zener model would be more convincing.

Our response:

The reviewer pointed out that the change of the Curie temperature (T_C) and spin split energy (ΔE) in the two devices A and B seem to be understood as a result of the increase of $N_0 \alpha$, since $T_C \sim (N_0 \alpha)^2 \sim \Delta E^2$ in the mean-field Zener model, if we neglect the difference in the Fe concentration x . However, as we explained in the Discussion section of the manuscript, it is the large discrepancy between the $N_0 \alpha$ values estimated from T_C and ΔE of the same sample that clearly indicates the failure of the mean-field Zener model for FMSs, at least in the case of (In,Fe)As. For example in device A, the $N_0 \alpha$ value estimated from ΔE (= 31.7 meV) is 0.21 eV, while that estimated from T_C (= 45K) is 4.5 eV, which differ by 20 times. In Fig. R3a, we show the experimental $\Delta E - T$ data and the theoretical predictions from the mean-field Zener (MFZ) model of device A (the experimental data are red circles, and the theoretical curves are green and blue curves, respectively). The two theoretical curves are calculated assuming the total angular momentum $J = 5/2$ for Fe^{3+} state. The green curve was calculated with the experimental value T_C of 42 K, which yields $N_0 \alpha = 4.5$ eV and $\Delta E = 675$ meV. The blue curve was calculated with the experimental value ΔE of 32 meV, corresponding to

$N_0\alpha = 0.21$ eV and $T_C = 0.093$ K. Both curves largely deviate from the experimental $\Delta E - T$ data. Therefore, the mean-field Zener model either underestimates T_C or overestimates ΔE by one or two orders of magnitude. Our results clearly indicate that the *absolute values* of T_C and ΔE cannot be described consistently by the mean-field Zener model, at least in the case of (In,Fe)As.

Fig. R3 . (a) Experimental spin split energy (ΔE) as a function of temperature T of device A (red) and the calculated curves by the mean-field Zener (MFZ) model (green curve and blue curve for $N_0\alpha = 4.5$ eV and 0.21 eV, respectively). Both axes are plotted in logarithmic scale. (b) $\Delta E - T$ data of device A (red) and device B (blue). Dotted curves are the Brillouin-function fittings using the experimental values of ΔE and T_C .

On the other hand, if we treat the ΔE and T_C values as separated parameters, not related to each other by the descriptions of the mean-field Zener model, we found that the Brillouin-function fittings are in considerably good agreement with the $\Delta E - T$ data of devices A and B. In Fig. R3b, we show the $\Delta E - T$ data of devices A and B and two Brillouin-function fitting curves (dotted red and blue curves). These two Brillouin-function curves are generated by $T_C = 42$ K and $\Delta E = 32$ meV (device A, dotted red curve) and $T_C = 65$ K and $\Delta E = 50$ meV (device B, dotted blue curve). The two dotted curves explain quite well the experimental data in devices A and B. These results indicate that although the mean-field Zener model proposed by T. Dielt *et al.* [*Phys. Rev. B* **63**, 195205 (2001)] fails, the magnetic properties of (In,Fe)As can be described quite satisfactorily by other mean-field approaches rather than the Zener model.

Corresponding revised parts:

- ✧ In page 7, line 153 in the revised manuscript, we added the comparison between the $\Delta E - T$ data of devices A and B with the Brillouin-function fitting curves: “We also show in Fig. 2e ... which will be discussed later in the Discussion section.”
- ✧ We revised Fig. 2e in the manuscript as shown in Fig. R3b, added error bars of the ΔE data and the two Brillouin fitting curves.

4) In Fig. 3c, the majority valley becomes smaller under larger magnetic field; in Fig. 3a, c and d, the majority valleys are smaller than minority valley, which is not explained. The plots making the comparison between the different field strengths is interesting, could the data from -1T to +1T be made available in the supplementary material.

Our response:

Fig. R4. (= Fig. 3a – d in the main manuscript) a – d. Evolution of the $d^2I/dV^2 - V$ curves with external magnetic fields. Experimental data (open circles) and their fitting curves (solid curves) are shown in the bottom panels, where white and black arrows indicate the minority and majority valley centers, respectively (The vertical axes are shifted for clear vision). The fitting Lorentzian curves (without the linear offset terms) for the data at 0T (black) and 1T (red) are shown in the top panels. a and b are the results of device A at 3.5 K and 50 K, c and d are the results of device B at 3.5 K and 50K, respectively.

In Fig. R4, we show the two-valley structures in the $d^2I/dV^2 - V$ curves of devices A (Figs. R4a and b) and B (Figs. R4c and d) under various magnetic fields H , at 3.5 K and 50 K (= Figs. 3a-d in the main manuscript). As a fitting function of the two-valley

structures, we used the sum of two Lorentzian curves of eq.(R5), as we will describe in page R20. Note that at the upper panels of Fig. R4a and b, we excluded the linear offset term ($BV+C$) in the fitting Lorentzian curves of eq.(R5) for clear analysis of the two spin valleys (all the terms of eq.(R5) are included in Fig. 3 of the main manuscript). From the upper panels of Figs. R4a, c, and d (the minority and majority spin Lorentzian fitting curves), one can see that in Figs. R4a and d, the majority spin valley is smaller (shallower) than the minority spin valley at zero magnetic field, whereas in Fig. R4c the majority spin valley became shallower with applying H , as commented by the reviewer. In the following, we explain possible reasons for this complicated behavior.

First, we show in Fig. R5a the $dI/dV - V$ curves of device B at 3.5 K and under 0 T (black) and 1 T (red). One can see that the shallowing of the majority spin valley in the $d^2I/dV^2 - V$ curve is caused by the increase of the dI/dV after the end of the direct tunnelling region ($V \sim V_{\text{major}}$) of the Esaki diode (indicated by the red arrow in Fig. R5a). As illustrated in Fig. R5b, at the end of the tunnelling region ($V \sim V_{\text{major}}$), the (In,Fe)As conduction band (CB) bottom (majority spin CB bottom) is lifted to the same energy as the p^+ -InAs valence band (VB) top, and direct tunnelling from CB to VB is suppressed. Therefore, the increase of the dI/dV after the end of the tunnelling region reflects the tunnelling conductance due to other *indirect* tunnelling processes, such as magnon-assisted tunnelling, phonon-assisted tunnelling, gap-state assisted tunneling, or their combinations.

In Fig. R5c, we plot $d^2I/dV^2 - V$ curves of device B, measured at 3.5 K under various magnetic fields H from -1 T to 1 T applied in the film plane (the data at 0 T and 1 T are the same as those plotted in Fig. R4c). To show the dependence of the asymmetry between the majority and minority spin's valleys on the magnetic field H , we plot in Fig. R5d the difference in the d^2I/dV^2 values at the majority and minority spin's valley center, $\Delta d^2I/dV^2 = d^2I/dV^2(V_{\text{major}}) - d^2I/dV^2(V_{\text{minor}})$, as a function of the magnetic field H . We see that $\Delta d^2I/dV^2 - H$ shows the same nonlinear behavior under positive and negative H . This result indicates that the indirect tunnelling process in device B is magnetic-field dependent.

Here, we propose a scenario of *gap-state assisted tunnelling through paramagnetic Fe-induced gap states* at the interface of the p-n junction: At the interface or in the depletion region of the p-n junction, some Fe gap states can exist due to diffusion of Fe atoms from the (In,Fe)As electrode. The energy levels of these paramagnetic Fe-induced states are close to the CB bottom of (In,Fe)As [see K. Huang et al., *J. Appl. Phys* **64**, 6770 (1988)]. Thus, electrons at the CB bottom of (In,Fe)As can indirectly tunnel to the VB top of p^+ -InAs through these paramagnetic Fe gap states after the end of the direct

tunneling region.

Fig. R5 (= Supplementary Fig. S3 in Supplementary Information). (a) $dI/dV - V$ curves of device B, measured at 3.5 K without and with a magnetic field H of 1 T applied in the film plane. At 1 T, dI/dV increases after the end of the tunnelling region ($V \sim V_{\text{major}}$), which causes the shallowing of the majority spin valley in the $d^2I/dV^2 - V$ curves at 1 T observed in the upper panel of Fig. R4c. (b) Schematic energy diagram of the p-n junction at $V \sim V_{\text{major}}$. The increase of dI/dV is due to indirect tunnelling processes through gap states at the interface. (c) $d^2I/dV^2 - V$ curves of device B at 3.5 K, measured under various magnetic fields H from -1 T to 1 T applied in the film plane. (d) Difference in the d^2I/dV^2 values of (c) at the majority and minority spin's valley centers, $\Delta d^2I/dV^2 = d^2I/dV^2(V_{\text{major}}) - d^2I/dV^2(V_{\text{minor}})$, as an indicator of the asymmetry between the two valleys under different H .

Figures R6 and R7 illustrate the schematic energy diagrams of the paramagnetic Fe gap states in devices A and B, respectively, at different temperatures and magnetic fields. Using these diagrams, we will explain the behavior of the two spin valleys in the $d^2I/dV^2 - V$ curves in Figs. R4a, c, and d, as follows.

Fig R6 (= Supplementary Fig. S4 in Supplementary Information). Schematic energy diagrams of the gap-state assisted tunnelling through paramagnetic Fe states (green lines) in device B. (a) and (b) are the situations at 3.5 K, without and with H , whereas (c) and (d) are the situations at 50 K, without and with H , respectively. At 3.5 K and 0 T (panel a) the gap-state assisted tunnelling is contributed mainly from the majority CB (blue arrow), whose energy is close to that of the Fe gap states. At 50 K and 0 T (panel c) both of the majority spin (blue arrow) and minority spin (red arrow) CBs contribute to the Fe gap-state assisted tunnelling. At 1 T (panels b and d), the spin angular momentums of the Fe gap states are aligned with the majority spin in the CB of (In,Fe)As, thus the tunnelling from the minority spin CB is prohibited, while the probability from the majority spin CB is enhanced.

• **Behavior of the $d^2I/dV^2 - V$ curves in Fig. R4c (device B, 3.5 K)**

At 3.5 K and 0 T (Fig. R6a), the Fe gap-state assisted tunnelling occurs mainly from the majority spin CB, whose energy is close to that of the Fe gap states, to the $p^+\text{-InAs}$ VB through the paramagnetic Fe gap states that have the same spin magnetic moment.

At 3.5 K and 1 T (Fig. R6b), however, there are more Fe gap states whose magnetic moments aligned with the majority spins in the CB of (In,Fe)As. Thus, indirect tunnelling from the majority spin CB is enhanced. This explains the increase of the dI/dV after the end of the direct tunneling region (Fig. R5a) and the shallowing of the majority spin valley of the $d^2I/dV^2 - V$ curves (Fig. R4c) when H was applied.

• **Behavior of the $d^2I/dV^2 - V$ curves in Fig. R4d (device B, 50 K)**

In the upper panel of Fig. R4d, the majority spin valley is shallower than the minority spin valley at 0 T, and becomes slightly deeper when applying H , although it is still shallower than the minority spin valley. We can explain this behavior as follows. At 50 K, electrons from both the majority spin (blue arrow) and minority spin (red arrow) CBs can tunnel through the paramagnetic Fe gap states and contribute to the indirect tunnelling current, as shown in Fig. R6c. This is possible because of the smaller spin split energy of (In,Fe)As CB and phonon-assisted processes existing at 50 K. This explains why the majority spin valley at 50 K (Fig. R4d) is shallower than that at 3.5 K (Fig. R4c) at 0 T. However, when H was applied, more magnetic moments of the paramagnetic Fe states are aligned with the majority spin in the CB of (In,Fe)As, and the indirect tunnelling from the minority spin CB is partly suppressed, as shown in Fig. R6d. Therefore the total indirect tunnelling current decreases, which explains why the majority spin valley of the $d^2I/dV^2 - V$ curve at 50 K in devices B becomes less shallow (slightly deeper) with applying H as seen in Fig. R4d.

• **Behavior of the $d^2I/dV^2 - V$ curves in Fig. R4a (device A, 3.5 K)**

In the upper panel of Fig. R4a (Fig. 3a in the main manuscript), the majority spin valley is shallower than the minority spin valley even at 0 T, and the two spin valleys change very little with H . Figure R7 shows the schematic energy diagrams of the gap-state assisted tunnelling through paramagnetic Fe states (green lines) in device A at bias voltages $V = V_{\text{major}}$ and $V = V_{\text{minor}}$. In device A, because the Fermi level of (In,Fe)As lies above the bottom of the majority spin CB but below that of the minority spin CB, the tunneling direction of electrons at $V = V_{\text{minor}}$ is opposite to that at $V = V_{\text{major}}$, as illustrated in Fig. R7 (please also refer to Fig. R1 of this Response Letter). At 0 T and $V = V_{\text{major}}$ (Fig. R7a) the Fe gap-state assisted tunnelling current is contributed only by electrons in the majority spin CB (blue arrow) of (In,Fe)As, because the minority spin CB (red arrow) is empty. Meanwhile, at $V = V_{\text{minor}}$ (Fig. R7b,d) the minority spin electrons in the VB of p^+ -InAs, however, cannot tunnel into the minority spin CB of (In,Fe)As through the Fe gap states because the energy levels of the Fe gap states are lower than the minority CB bottom edge. Therefore the Fe gap-state assisted tunnelling current at $V = V_{\text{minor}}$ is zero. This difference of the Fe gap-state assisted tunnelling currents in the cases of $V = V_{\text{major}}$ and $V = V_{\text{minor}}$ explains why the majority spin valley is shallower than the minority spin valley at 0 T. We also note that because the electron density n of (In,Fe)As is lower ($\sim 1 \times 10^{18} \text{ cm}^{-3}$) than device B, the depletion layer of the p-n junction extends more into the (In,Fe)As side. Due to the lack of carriers inside the

depletion region of (In,Fe)As, more Fe atoms act as paramagnetic Fe states, which increases the number of the Fe gap states. This situation further enhances the Fe gap-state assisted tunnelling current at $V = V_{\text{major}}$ in device A in comparison with that of device B.

When applying $H = 1$ T (Fig. R7c), the number of majority spin Fe gap states increases. However, due to the small electron density n in the CB of (In,Fe)As layer, an increase in the number of majority spin Fe gap states (which is already quite large at 0 T) does not yield any large effect. This is why the gap-state assisted tunnelling current shows almost no change. Besides, other magnetic-field-independent mechanism (ex. phonon-assisted tunnelling) may be dominant in device A.

Fig R7 (= Supplementary Figure S5 in Supplementary Information). Schematic energy diagram of the gap-state assisted tunnelling through paramagnetic Fe states (green lines) in device A at 3.5 K. (a) and (b) are the situations at 0 T, at V_{major} and V_{minor} , whereas (c) and (d) are the situations at 1 T, at V_{major} and V_{minor} , respectively. Due to the smaller electron density n of (In,Fe)As in device A, the depletion layer of the p-n junction extends deeper into the (In,Fe)As side, which increases the number of Fe gap states. Because the minority spin CB of (In,Fe)As (red arrow) in device A is empty, the gap-state assisted tunnelling is contributed only by the majority spin CB (blue arrow). At 1T (panel c, d) the spin angular momentums of the Fe gap states are aligned with the majority spin in the CB of (In,Fe)As. However due to small n of the (In,Fe)As layer, the indirect tunnelling current from the majority spin CB is almost unchanged.

Corresponding revised parts:

- ✧ We added Fig R5, R6, and R7 (as Supplementary Figure S3, S4, and S5), and our explanation of the behaviors of the two spin valleys in the $d^2I/dV^2 - V$ curves of the two devices A and B in Supplementary Information as Supplementary Note 3.

5) In Fig. 3e the magnetic field dependence of ΔE for device A is in a paramagnetic state at 50 K, why is the linear behavior of the Zeeman effect is not observed?

Our response:

The Curie temperature (T_C) of the (In,Fe)As layer in device A is 42 ~ 45 K, estimated from the temperature dependence of the spin split energy ΔE (Fig. R2 or Fig. 2e in the revised manuscript) and the magnetization M (Supplementary Fig. S1b in Supplementary Information). The measurement temperature of 50 K is thus very close to T_C of the sample. In this temperature range, although the global ferromagnetic order disappears, the local ferromagnetic order can still remain due to the fluctuation in the local Fe concentration or local electron density. It is noteworthy that for (In,Fe)As thin films without co-doping donors such as Si or Be, the typical electron density is around $2 \times 10^{18} \text{ cm}^{-3}$, which is very close to the vicinity of metal-insulator transition (MIT) [see P. N. Hai et al., *Appl. Phys. Lett.* **101**, 182403 (2012)]. In this situation, quantum critical fluctuation in the local carrier density is likely to occur as observed in (Ga,Mn)As [see M. Sawicki et al., *Nature Phys.* **6**, 22-25 (2010)]. Therefore, the $\Delta E - H$ curve of device A at 50 K does not show the linear behaviour of the Zeeman effect, but shows superparamagnetic-like behaviour.

Corresponding revised parts:

- ✧ In page 8 line 178, we changed “*giant g-factor*” to “*giant effective g-factor*”.
- ✧ We added a short explanation of the behavior of the $\Delta E - H$ curve of device A at 50 K in the revised manuscript, page 8, line 179: “*We note that the measurement temperature of 50 K is very close to T_C of device A. ...effectively enhance the g-factor.*”

6) The section dealing with the analysis of the "Magnetic anisotropy of the band structure of (In,Fe)As" need a bit more care. Could the difference in the curves of +1T and -1T fields be due to sample alignments?

Our response:

The difference in the $dI/dV - V$ curves at 1T and -1T is caused by an *odd* function contribution of the magnetic field H , due to the Hall effect in the p^+ -InAs substrate, as

explained below.

The devices were placed meticulously in the center position of the space between the two poles of our electromagnet (misalignment, if any, should be in millimeter order). In principle, with the center position as the origin, the distribution of magnetic field in this sample space is an even function of the position. Figure R8a shows the position dependence of H in the sample space of our electromagnet measured by a Gaussmeter, which confirmed the even-function symmetry of the magnetic field. Therefore misalignment of the sample from the center position, if any, cannot generate a response that is an odd function of H .

Fig. R8 (= Supplementary Figure S6 in Supplementary Information). (a) Magnetic field in the sample space between two poles of the electromagnet measured by a Gaussmeter, which shows even-function symmetry of the magnetic field around the center position. (b) Schematic sketch of the diode device placed in the magnetic field H_X , which is along the X direction. Red lines illustrate the current paths, which are mainly in the Z direction. However, in the p^+ -InAs substrate (yellow areas) currents flow in the X-Y plane. The magnetic field H_X and the currents with Y-direction components induce Hall voltages in the Z direction. These Hall voltages, which are odd functions of the magnetic field, are not canceled out due to different path lengths of the currents in the substrate, thus can be detected by the top and bottom electrodes.

In Fig. R8b, we show a general situation when the sample is placed in the X-Y plane under a magnetic field H applied in the X direction (H_X). The currents (red lines) flow

through the mesa diode in the Z direction. In the thick p⁺-InAs substrate, however, the currents can flow in the X-Y plane (yellow areas in Fig. R8b). Because the top mesa is not located exactly at the center of the substrate, different path lengths are expected for currents flowing in different directions in the X-Y plane (see the top-view in Fig. R8b). In these yellow areas, the magnetic field H_x and the currents in the Y-direction induce Hall voltages in the Z direction. These Hall voltages are not canceled out due to different path lengths of the currents in the substrate, thus can be detected by the top and bottom electrodes. These Hall voltages are odd functions of the magnetic field and thus have opposite signs at +1T and -1T. We think that this is the main origin of the difference in the $dI/dV - V$ data under +1T and -1T.

Corresponding revised parts:

We added Fig R8 (as Supplementary Figure S6), and our explanation of the origin of the asymmetric behavior in the $dI/dV - V$ curves of device A measured at +1T and -1T in Supplementary Information as Supplementary Note 4.

(Fig. 4b) The reason for fitting the experimental data is rather poorly introduced and seems to be more justified after consideration of facts.

Our response:

The TAMR results in Fig. 4a in the main manuscript are divided into two regions; the tunnelling region (top panel, bias voltage $V = -0.10 \sim 0.05$ V) and the diffusion region (bottom panel, $V = 0.42 \sim 0.49$ V).

The results in the tunnelling region show a four-fold symmetry and another higher-order (eight-fold) symmetry. The four-fold symmetry reflects the cubic symmetry of the zinc-blende crystal structure of (In,Fe)As. Meanwhile, the eight-fold symmetry has been observed in the crystalline anisotropy magnetoresistance (AMR) of (In,Fe)As [see P. N. Hai et al., *Appl. Phys. Lett.* **100**, 262409 (2012)]. Therefore we fitted these data by a four-fold term and an eight-fold term.

On the other hand, the data in the diffusion region are dominated by two-fold terms. The symmetry axis of the two-fold symmetry in the diffusion region rotates by 45 degrees (from [010] to [110]) as the bias voltage V increases from 0.42 to 0.49 V. This indicates that there are at least two two-fold terms with different symmetry axes in this region. Therefore, we fitted the data in this region by two two-fold terms (with the symmetry axes along [110] and [010]) and a four-fold term which is typical of zinc blende structure.

Corresponding revised parts:

✧ In page 10, line 227 of the revised manuscript, we added the explanation of the

fitting process: “One can see the TAMR results in ...symmetry axes in this region.”

Curiously Fig 4(b) 1 which shows the cross sectional from Fig.4(a) data at 50mV, in the tunneling region of sample A, the raw data does not appear to be as symmetric as the curve fit. Is there a plausible explanation for the asymmetric behavior of the peaks?

Our response:

As described above and in Fig. 4b-1, in the tunnelling region the $\Delta(dI/dV)$ signal shows a four-fold symmetry and another higher-order (eight-fold) symmetry. The asymmetric behavior of the peaks in Fig. 4b-1 is quite weak and changes randomly at different bias voltages in the tunnelling region. We think that this asymmetry is due to the measurement noise and does not represent any meaningful physics.

Responses to reviewer #2:

1) Is the (In,Fe)As n^{++} or n-type? I think this distinction is important.

Our response:

Although the electron density n of the (In,Fe)As layers in the two samples cannot be measured by Hall effect measurements due to the parallel conduction in the conducting p^+ type InAs substrates, the typical n values for (In,Fe)As samples with and without Be co-doping are in the order of 1×10^{19} and $1 \times 10^{18} \text{ cm}^{-3}$, respectively [see P. N. Hai et al., *Appl. Phys. Lett.* **101**, 252410 (2012); P. N. Hai et al., *Appl. Phys. Lett.* **101**, 182403 (2012)]. Therefore, (In,Fe)As thin films in this work are n^+ type.

Corresponding revised parts:

- ✧ In page 4, line 81 of the revised manuscript, we described the rough estimation of the electron density in our two samples: “*Although the electron density of the (In,Fe)As layers in ... are in the order of 1×10^{19} and $1 \times 10^{18} \text{ cm}^{-3}$, respectively^{19,22,23}.*”
- ✧ We revised the notation n-(In,Fe)As to n^+ -(In,Fe)As and p-InAs to p^+ -InAs in several places in the revised manuscript.

2) Abstract: authors mention that weak nature of s-d exchange is a "common belief". I have to ask them to reword this statement, because s-d exchange is well measured in II-VI and III-V semiconductors over the last few decades, so it is not a belief, but a result of many measurements.

Our response:

Following the reviewer’s comment, we have revised the abstract, in which we

mentioned the experimental results of the *s-d* exchange interaction in zinc blende semiconductors.

Corresponding revised parts:

- ✧ In page 1, line 20, we rewrote in the abstract: “...*which is surprising considering the very weak s-d exchange interaction that has been reported in several zinc-blende (ZB) type semiconductors*^{16,17}”.
- ✧ We cited ref. 16 [Furdyna, J. K., Diluted magnetic semiconductors. *J. Appl. Phys.*, **64**, R29 (1988)], which is a review paper on II-VI DMSs and contains various $N_0\alpha$ values of these materials, and ref. 17 which estimated the $N_0\alpha$ value in (Ga,Mn)As.

3) Page 2, line 39, the authors cited Kobayashi et al for the proof of the Fermi level position in the impurity band of Mn. I recommend in addition they cite Chapler et al PRB 87, 205314 (2013) because that work shows the clear presence of impurity band and Fermi level in GaMnAs.

Our response:

Corresponding revised parts:

We added and cited the paper of Chapler et al. as Ref. 15 in the revised manuscript.

4) Page 2, line 44, the authors mention that the lack of observation of valence band spin splitting casts doubt on the validity of the mean-field Zener model. I think this is too strongly worded because the mean-field Zener model does accurately predict the ferromagnetism in semiconductors, however the measurements indicate that it does not apply to some materials, so its range applicability is narrower than previously thought.

Our response:

Corresponding revised parts:

- ✧ We rewrote the sentence in page 2, line 48 as: “*These casted doubts on the validity of the mean-field Zener model as a **universal** model to describe and predict the magnetic properties of FMSs.*”

5) Page 3, line 74, can the authors also please include the electron density (n) for each sample?

Our response:

As explained in the response no.1), although the electron density n of the (In,Fe)As layers in the two samples cannot be estimated by Hall effect measurements due to the parallel conduction in the p^+ type InAs substrates, the typical n values for (In,Fe)As

samples with and without Be co-doping are in the order of 1×10^{19} and $1 \times 10^{18} \text{ cm}^{-3}$ [see P. N. Hai et al., Appl. Phys. Lett. **101**, 252410 (2012); P. N. Hai et al., Appl. Phys. Lett. **101**, 182403 (2012)].

Corresponding revised parts:

✧ In page 4, line 81 of the revised manuscript, we described the electron density in our two samples: “Although the electron density of the (In,Fe)As layers in ... are in the order of 1×10^{19} and $1 \times 10^{18} \text{ cm}^{-3}$, respectively^{19,22,23}.”

6) Page 4, lines 83-86, authors explain tunneling region, however they don't explain why the tunneling has a spin dependence (important for general Nature audience).

9) Page 5, lines 106-109, authors are explaining how the I-V behavior can probe the spin-split DOS, however there is not much explanation here. For general audience of Nature, it is important to explain this concept. Please provide a cartoon to explain how dI/dV probes DOS and spin-split DOS.

Our response:

Following the reviewer's comments and suggestions, we have revised several parts in the explanation of the experimental principles and methods, as follows:

Fig. R9 (= Fig. 1c in the revised manuscript). Illustration of the band structures of n^+ -(In,Fe)As and p^+ -InAs, and the $dI/dV - V$ curves of the Esaki diode at temperatures above (left panel) and below (right panel) T_C . In principle, the spin splitting in DOS of the (In,Fe)As CB leads to the “kink” behavior (indicated by two black arrows) in the $dI/dV - V$ curve at low temperature ($T < T_C$).

Corresponding revised parts:

We explained the reason why the spin-split DOS can be probed using $I-V$ measurement

results in the tunnelling region.

- ✧ In page 4, line 96, we added: “*Because the tunnelling conductance is proportional to the ...and temperature dependence of the “kink” structure.*”
- ✧ We added schematic viewgraphs that explain the change in DOS of the (In,Fe)As CB and the corresponding change in $dI/dV - V$ curves at $T > T_C$ and $T < T_C$, as shown in Fig. R9 (the same as Fig. 1c of the revised main manuscript).

7) Page 4, lines 88-89, the authors mention that there should be no current in the band gap region, however, they are not mentioning the possibility of thermionic current from electrons and holes thermally hopping and diffusing over the barrier. Their explanation is only valid at 0K.

Our response:

Although there is contribution of the thermionic current, we think that the contribution of the paramagnetic Fe gap-state assisted tunneling is more important, because we can explain the experimental magnetic-field and temperature dependence of the majority and minority spin valleys, as discussed in our responses to the comment 4) of the reviewer 1. We added comments on the origins of the tunneling current in the band-gap region of our devices.

Corresponding revised parts:

- ✧ In page 5, line 109 of the revised manuscript, we rewrote: “*At $e^{-1}(E_n + E_p) < V < e^{-1}E_g$, corresponding to region ② (bandgap region), although the tunnelling of electrons... tunnelling of electrons through the gap states, which usually exist in heavily-doped semiconductors.*”

8) Page 4, line 92, authors state that diffusion region occurs once the applied bias is larger than the built-in bias, however again there is some mistake in this explanation. At finite temperature, electrons and holes undergo diffusive forward bias current due to thermal diffusion over the barrier. At 0K, diffusion current only occurs at $V_{\text{builtin}} = V_{\text{applied}}$, i.e. flat band. Once $V_{\text{applied}} > V_{\text{builtin}}$, then the current is no longer diffusive, but changes to majority drift.

Corresponding revised parts:

We revised the explanation of the diffusion region.

- ✧ In page 5, line 115 :” *Finally at larger bias voltages ($e^{-1}E_g < V$, corresponding to region ③), the occupied states in the CB (or VB) of n^+ -(In,Fe)As reach the same*

energies as the unoccupied states in the CB (or VB) of p^+ -InAs, **diffusive and drift currents start to flow as in normal diodes. Thus we call this region ③ diffusion region.**”

10) Page 5, lines 117-118, the authors conclude that the two-valley structure corresponds to the spin-splitting of the CB. This is the central premise of the study, and I fully agree with this conclusion.

Our response:

We thank the reviewer for his encouraging comment.

11) Page 5, lines 121-123, the authors discuss that the two valleys are fitted to extract the spin splitting ΔE , however they don't provide the fit function. They should at a minimum provide the fit function, explain its physical justification, how many fit parameters are used, and what is the uncertainty of the extracted parameters, i.e. error bars. This is especially important for the data near T_C , where clearly the broadening of the peaks means that the fit uncertainty must necessarily diverge.

Our response:

To analyze the two-valley features of the $d^2I/dV^2 - V$ curves, we used the following fitting function, which is the sum of two Lorentzian curves and a linear offset:

$$\frac{d^2I}{dV^2} = A_{\text{minor}} \frac{\Delta_{\text{minor}}/2}{(V - V_{\text{minor}})^2 + (\Delta_{\text{minor}}/2)^2} + A_{\text{major}} \frac{\Delta_{\text{major}}/2}{(V - V_{\text{major}})^2 + (\Delta_{\text{major}}/2)^2} + BV + C \quad (\text{R5})$$

Here, A_{minor} (A_{major}) is the magnitude, Δ_{minor} (Δ_{major}) is the full width at half maximum, V_{minor} (V_{major}) is the center bias voltage of the valley corresponding to the minority (majority) spin CB, respectively, and $BV + C$ is the linear offset. We note that the linear slope B is needed only in the case of device A, because the $d^2I/dV^2 - V$ curves often have a linear offset in the vicinity of zero bias. In device B, the B parameter is set to be 0. The valley center's positions V_{major} and V_{minor} correspond to the band edges of the majority and minority spin CBs, respectively. The standard errors of the fitting results for V_{major} and V_{minor} , which are estimated using the MATHEMATICA software, are very small (< 0.5 mV), indicating very good agreement between the experimental data and the fitting curves. We defined the errors in the estimation of the ΔE data by summing the standard errors of V_{major} and V_{minor} , and added error bars in Fig. 2e and Figs. 3e and f of the main manuscript. Except for a single data of device A at 40 K, all other data points have the error bars smaller than the data point's size.

Corresponding revised parts:

- ✧ In page 6, line 138 of the revised manuscript, we added the explanation of the fitting process as: *“To analyse these two-valley structures, we use ...which is smaller than 1 meV in almost all the data points.”*
- ✧ We revised Fig. 2e, Fig. 3e, and Fig. 3f, and added error bars to the ΔE data points.

12) Page 6, lines 124-125, the authors state that the spin splitting of device B is larger than A because of the higher Fe concentration. However, they don't mention what theory or model they base that prediction on. The authors say that mean-field Zener doesn't apply, yet they still want to say that the T_C scales with Fe doping. They should explain what theory supports their statement.

Our response:

While the mean-field Zener model fails to explain the high T_C value in our case by a factor of 100, the $\Delta E - T$ experimental data follow the Brillouin-function fitting, indicating that the magnetic properties of (In,Fe)As can be described quite satisfactorily by other mean-field approaches rather than the Zener model (Please also see Fig. R3 and our responses to the comment 3) of reviewer 1). Within a mean-field theory framework, as long as there are $s-d$ exchange interactions in the system, we think it is quite natural to expect that the spin split energy ΔE increases with increasing the Fe concentration x , irrespective of the Zener model. This is because the more Fe doping into the lattice leads to the higher probability that free electrons can interact with Fe spins.

On the other hand, from the experimental results, at 3.5 K the ratio $\Delta E(\text{device A}) : \Delta E(\text{device B}) = 50 : 32 = 1.6$, which is larger than the ratio $x(\text{device A}) : x(\text{device B}) = 8 : 6 = 1.33$. This indicates that the experimental values of ΔE (at 0 K) do not simply increase proportionally with x in both devices, which deviates from the theoretical prediction of the mean-field Zener model.

Corresponding revised parts:

- ✧ In page 7, line 153 in the revised manuscript we added the discussion on the comparison of the experimental $\Delta E - T$ data of devices A and B with the mean-field theories: *“We also show in Fig. 2e the two Brillouin-function fitting curves ... which will be discussed later in the Discussion section.”*

13) Page 6, lines 135-141, the authors explain the magnetic field dependence of the spin splitting device A at 50K, and they claim that the data indicate a g-factor of 621, which is a giant g-factor induced in InFeAs which is larger than observed in II-VI

DMSs. Here I actually strongly disagree with the authors. Their M vs T data are too difficult to fit the T_c accurately, therefore they don't really know if all ferromagnetism in sample A is gone by 50K. In fact, there is plenty of spontaneous magnetization at 50K in their data. Thus the anomalously large g-factor is likely just because 50K isn't high enough to make that sample exhibit pure paramagnetism. Only if the authors did the field dependence at higher temperature (where there is no M_{spont}) and fitted with Brillouin function could this be believable. I am not convinced that InFeAs shows anomalously large s-d exchange based on the data shown.

Our response:

We agree with the reviewer that the temperature of 50 K is not high enough to make the (In,Fe)As thin film in device A exhibit pure paramagnetism. However, measurements at temperatures much higher than T_c , which are required to accurately estimate the g-factor of paramagnetic (In,Fe)As, are very difficult because of the broadening of the tunnelling spectroscopy at high temperatures. Thus, the g-factor at 50 K is considered as an “effective” g-factor. We made a comment on this and deleted the comparison of the g-factor value in (In,Fe)As with the values in II-VI DMSs in the main manuscript.

Corresponding revised parts:

- ✧ In page 8 line 178, we changed the phrase “*giant g-factor*” to “*giant effective g-factor*”.
- ✧ In page 8, line 179 of the revised manuscript, we added a comment: “*We note that the measurement temperature of 50 K is very close to T_c ...is difficult because of the broadening of the tunnelling spectroscopy at high temperatures*”
- ✧ We deleted the following sentence in the previous manuscript: “*which is larger than that observed in II-VI diluted magnetic semiconductors (DMSs)²¹.*”

14) Page 6, lines 145-147, authors state that there is still room to increase T_c in InFeAs, but this rather vague. The authors should be more specific and say how much more room there is to increase Fe doping or increase the n-type doping. Specifically, what are the defect limits? Solubility limits?

Our response:

We expect that the Curie temperature (T_c) of (In,Fe)As can be largely increased by increasing either the Fe concentration x or the electron density n , as is commonly observed in carrier-induced ferromagnetic semiconductors. Since the Fe concentration x (6 and 8%) and electron density n ($\sim 1 \times 10^{19} \text{ cm}^{-3}$) in the present (In,Fe)As samples are still far below the maximum values achieved in Mn-doped III-V FMSs (the maximum

Mn doping concentration is ~20% and the maximum hole density is $\sim 10^{21} \text{ cm}^{-3}$), there is still much room for increasing either x or n , and consequently T_C in (In,Fe)As. The highest x that has been reported so far for (In,Fe)As is 9% [see Hai, P. N. *et al.*, *Appl. Phys. Lett.* **101**, 182403 (2012)], but this can be increased by optimizing the growth conditions or using special techniques such as delta doping. On the other hand, the control of n by chemical doping so far has been limited only to the use of Be or Si. Although Be or Si atoms are doped in (In,Fe)As, n is limited to at most $1 \times 10^{19} \text{ cm}^{-3}$ due to their amphoteric behavior and low activation rates in InAs, especially at low growth temperature. Searching for good donors, possibly by using group VI elements such as Te or increasing n by electrical gating, are intriguing methods that may increase n to the order of 10^{20} cm^{-3} .

Corresponding revised parts:

✧ In page 8, line 191 of the revised manuscript, we revised our statement on the possibility of increasing T_C in (In,Fe)As: “*The temperature range where large spin-split energy in ...that may increase n to the order of 10^{20} cm^{-3} .*”

15) Page 8, lines 183-187, the authors conclude that the 4-fold symmetry in the tunneling region is much smaller than observed in the diffusion region because SOI is weaker than in the VB. But this conclusion is contradictory to the authors' main conclusion, which is that s-d exchange is larger than p-d exchange, however here they are stating the opposite. To be more specific in this section, the authors are stating that the VB spin splitting is larger than the CB splitting, which necessarily requires that p-d exchange is greater than s-d. However, their main conclusion is that p-d exchange = 0, and s-d is anomalously large. This contradiction needs to be resolved.

Our response:

In this paper we discussed about the *s-d* exchange interaction energy and its relation to T_C , but made no comment on the *p-d* exchange interaction. Actually, from the experimental data (the *I-V* curves) of our Esaki diodes, we were able to obtain neither the information about the spin-dependent structure of the valence band of (In,Fe)As nor the *p-d* exchange interaction energy in this material. Therefore we would like to leave the discussion on the *p-d* exchange interaction of (In,Fe)As open for future studies. We think that it is possible (and natural) that the *p-d* exchange interaction is even larger than the *s-d* exchange interaction in (In,Fe)As. Therefore there is no contradiction in our conclusions.

Corresponding revised parts:

✧ In page 4, line 90 of the revised main manuscript, we added a comment: “(we do

not take into account spin splitting of the VB of (In,Fe)As, because it is away from the Fermi level thus irrelevant to the present study)".

16) Page 9, lines 201-202, the authors state that s-d exchange is generally considered to be very weak. The wording should be changed, the s-d exchange is well measured in a wide range of DMSs, so it is not considered to be any value, it is just well measured. Rather than using such unqualified comparisons, the authors should directly state, we measure XXX s-d exchange which is different than the range of s-d exchange XX splitting observed in all other DMSs.

17) Page 9, lines 204-205, the authors should note that the $N_0\alpha$ values they observed are basically identical to the values observed in all II-VI DMSs. This means that the s-d exchange they measure is not surprisingly large, but actually exactly what would be expected based on measurements in II-VI DMSs carried out a few decades ago.

Our response:

As described in page R5-R6, the $N_0\alpha$ value that we estimated using the mean-field Zener model largely varies depending on which equation (Eq. (1) or Eq. (2)) is used: For example in device A, the $N_0\alpha$ value estimated from ΔE (= 32 meV) is 0.21 eV, while that estimated from T_C (= 45K) is 4.5 eV, which differ by one order of magnitude. This is because the mean-field Zener model is not applicable to (In,Fe)As. Thus, we now consider that any quantitative comparison of the $N_0\alpha$ values between (In,Fe)As and II-VI DMSs is inappropriate and would like to make no comparison. This revision does not affect the main conclusions of our paper.

We deleted our statement in the main manuscript in p.12: "*which is highly surprising because the s-d exchange interaction is generally considered to be very weak in ZB-type semiconductors*".

Corresponding revised parts:

✧ We deleted the following statement in page 12 in the main manuscript: "*which is highly surprising because the s-d exchange interaction is generally considered to be very weak in ZB-type semiconductors*".

18) Page 9, lines 213-214, the authors state that the measured s-d exchange splitting is not large enough to account for the observed T_c based on the mean-field Zener model. I agree with this statement. However, I would suggest that rather than stating that the Zener model is a failure (when in fact it explains magnetism in a number of DMSs), they can state that InFeAs in fact have a different physical mechanism for the

magnetism, which the Zener model does not capture.

Our response:

As the reviewer suggested, we have revised the statement about the failure of the mean-field Zener model. One of our conclusions is that the mean-field Zener model cannot explain the magnetic properties of (In,Fe)As and some other FMSs, such as high- T_C narrow-gap FMS (Ga,Fe)Sb reported recently [see N. T. Tu et al., *Appl. Phys. Lett.* **108**, 192401 (2016)]. This indicates that the mean-field Zener model is not a *universal* model for FMSs, and searching for other appropriate unified model for FMSs is strongly required.

Corresponding revised parts:

✧ We revised the Discussion section, from page 13, line 285 in the main manuscript: “*Besides the case of (In,Fe)As, the breakdown of the mean-field Zener model ...thus remains an unsolved theoretical challenge.*”, and cited works of (In,Mn)As, (In,Mn)Sb (grown by MOVPE) (ref. 37-39), and other theoretical papers that discuss the breakdown of the mean-field Zener model (ref. 31-33).

19) Page 10, lines 230-232, the authors are explaining how Be doping is used as both a donor and acceptor. This explanation needs to be a bit more detailed because of the general Nature audience. They need to explain that Be is amphoteric and can sit on either In or As site depending on the substrate temperature and therefore lead to either n or p type doping.

Our response:

Corresponding revised parts:

✧ In page 14, line 310 of the main manuscript, we have revised the explanation of the role of Be atoms in the Methods section as follows: “*In sample B we co-doped Be in the (In,Fe)As layer...because they favorably sit in the interstitial sites.*”

20) Fig2a-d, why are the vertical axes arbitrary units? In Fig. 1 the authors have dI/dV in units of mA/V, therefore d^2I/dV^2 should be units of mA/V².

21) Fig2e, the ΔE values need to have error bars to reflect the uncertainty in the fitting procedure.

Our response:

Corresponding revised parts:

✧ We have revised Fig. 2; we added units in the vertical axes of Fig. 2a-d, and added error bars of the ΔE data in Fig. 2e. In most of the data points, the error bars are smaller than the data point's size.

22) Fig. 3a-d, the vertical axis now has units of A/V^2 , however the axis is not labeled, so we can't actually know what size of the signal. They should include axes labels. Also, the plotted fitting curves don't appear to be offset Lorentzians, but more complex functions. It is appropriate for the authors to include the fit function in the manuscript.

Our response:

Corresponding revised parts:

- ✧ We have revised Fig. 3; we added labels in the vertical axes of Fig. 3a-d, and error bars in Fig. 3e,f. In most of the data points, the error bars are smaller than the point's size.
- ✧ In page 6, line 138 of the revised manuscript, we added the explanation of the fitting process as follows: “*We fitted these two-valley features by two Lorentzian curves ...which is smaller than 1 meV in almost all the data points.*”

23) Fig. 4c, if authors carried out fitting, the data points should have error bars to show the uncertainty of the fit parameters.

Our response:

We conducted the fitting of the $\Delta \left(\frac{dI}{dV} \right)$ data at various bias voltage V in the diffusion region. The fitting allows us estimate the anisotropy constants $C_{4[100]}$, $C_{2[010]}$, and $C_{2[110]}$, whose values were summarized in Fig. 4c. The standard errors of the anisotropy constant $C_{4[100]}$, $C_{2[010]}$, and $C_{2[110]}$ were estimated to be at most 0.055 (%) for all bias voltages. With this standard errors, the error bars in Fig. 4c are smaller than the size of the data points and cannot be seen.

Corresponding revised parts:

- ✧ We have revised Fig. 4, added error bars in Fig. 4c. In most of the data points, the error bars are even smaller than the point's size.

Responses to reviewer#3:

1) The reviewer concludes that either the theory is wrong or the measurements are wrong or possibly both. The use of tunneling spectroscopy data is fraught with experimental issues. First the band structures of both the conduction band and valence band are needed. The authors assume that the valence and conduction bands are parabolic. They ignore the contributions to the density of states from the heavy hole

band and light hole band in the valence band. Note k.p calculations of the valence band of InAs and InMnAs have been published see M. A. Meeker et al PRB 2015. Some discussion is needed regarding tunneling and band structure. Error analysis of tunneling spectroscopy data is needed. Also iron in InAs may have several charge states. The authors should discuss this since it may modify the tunneling spectra.

Our response:

We agree with the reviewer that we have to pay attention to the band structure of all the components in the analysis of the tunnelling spectroscopy data. In the following, we would like to show that, by carefully analyzing the $dI/dV - V$ characteristics of the Esaki diodes at various temperatures and magnetic fields, we can conclude that the contribution of the valence band (heavy hole, light hole) of p^+ -InAs does not affect the main conclusions of this paper.

Figure R10 shows the schematic band structures of $(\text{In,Fe})\text{As}$ and p^+ -InAs of our Esaki diodes at bias voltages V in the tunnelling region. As can be seen in Figs. 1d,e of the main manuscript, the tunnelling region in our devices A and B ends at $V = 60$ mV and 180 mV, respectively. Because the Fermi level E_F is 10 ~ 150 meV above the CB bottom in $(\text{In,Fe})\text{As}$ [see P. N. Hai et al., *Appl. Phys. Lett.* **101**, 252410 (2012)] and 8 meV below the VB top in p^+ -InAs, the bands participating to the tunnelling current are limited to the following:

Fig. R10. Band structures of $n^+(\text{In,Fe})\text{As}$ and p^+ -InAs of our Esaki diodes at bias voltages in the tunnelling region. The tunnelling from the Fe IB to p^+ -InAs VB is prohibited by the difference in orbital symmetry.

The electron-filled CB bottom and the possible Fe impurity band (IB) in the $n^+(\text{In,Fe})\text{As}$ side, and the empty VB top (the heavy hole (HH) and light hole (LH) bands) in the p^+ -InAs side. Thus the $dI/dV - V$ curves in this tunnelling region reflect the DOSs of these bands at various temperatures and magnetic fields.

For the p^+ -InAs VB, we have to consider the following two things:

- Because the p^+ -InAs is non-magnetic, the magnetic field dependence of the HH and LH bands should be very weak (the Zeeman energy of InAs at 1 T is smaller than 1 meV).

- Although the HH and LH bands can split with strain at low temperature, the strain-induced splitting is also very small because the p⁺-InAs layer was grown on the lattice-matched InAs substrate with very little or no strain. Furthermore, the strain-induced splitting, if any, would not show Brillouin-function-like temperature dependence or magnetic field dependence, contrasting to the spin splitting observed in the $d^2I/dV^2 - V$ curves in Fig. 2a-d and Fig. 3a-f in the main manuscript.

Therefore, from the magnetic field and temperature dependence of the $d^2I/dV^2 - V$ curves in Fig. 2 and Fig.3, we conclude that the change in the DOS of the VB of p⁺-InAs is *not* the origin of the observed experimental results.

Furthermore, in the n⁺-(In,Fe)As side, we have to consider two band components; the CB and the Fe-related IB.

- The Fe-related IB may show temperature and magnetic field dependence, but the spin split energy (~ 2 eV) of the *d*-band should be much larger than the observed value (32 \sim 50 meV).
- From the TAMR results shown in Fig. 4 of the main manuscript, the tunnelling from the Fe-related IB to the VB of p⁺-InAs seems to be prohibited by the difference in orbital symmetry. The $dI/dV - V$ data in the tunnelling region show only very weak four-fold symmetry and eight-fold symmetry of the CB of (In,Fe)As (Fig. 4a and Fig. 4b1 of the main manuscript), while in the diffusion region we observed strong two-fold symmetries (Fig. 4a and Fig. 4b 2-6 of the main manuscript) that are supposed to originate from the Fe-related IB.

Therefore, by investigating the temperature and magnetic field dependence, we concluded that the splitting behavior in the $d^2I/dV^2 - V$ curves presented in this work is caused by the spin splitting in the CB of (In,Fe)As.

Next we discuss the charge states of Fe in (In,Fe)As. Although we did not conduct any direct measurement of the charge states of Fe, most of the Fe atoms are found to replace In sites in the neutral Fe³⁺ state. There are two reasons for this assignment; we have observed in (In,Fe)As 1) weak dependence of the electron density on the Fe concentration, and 2) weak temperature dependence of the electron mobility which indicates that the scattering by neutral impurities is dominant [See our previous study: Hai, P. N., *et al.*, *Appl. Phys. Lett.* **101**, 182403 (2012)]. As mentioned above, the tunnelling of electrons from the Fe-related IB into the VB of p⁺-InAs seems to be prohibited by orbital symmetry. Therefore the position of the Fe-related IB (which depends on the charge state of Fe) is not relevant to the analysis of the tunnelling spectroscopy in the tunnelling region of our diode devices. Note that in the diffusion region, where the transport of electrons from the Fe-related IB is no longer prohibited,

the TAMR results (Fig. 4 in the main manuscript) indicate that the Fe-related IB is close to the CB bottom or/and the VB top of (In,Fe)As, as described in the main manuscript (p.11-12). These results are consistent with the positions of the Fe^{3+} and Fe^{2+} charge states in InAs reported by Huang and Wessels [*J. Appl. Phys* **64**, 6770 (1988)].

Corresponding revised parts:

- ✧ In page 4, line 96 in the revised manuscript, we explained that the contribution from the VB of p^+ -InAs, if any, can be distinguished by measuring the temperature and magnetic field dependence of the d^2I/dV^2 - V curves: *“Because the tunnelling conductance is proportional to ... by investigating the magnetic field and temperature dependence of the “kink” structure.”*
- ✧ In page 8, line 186, we excluded the VB of p^+ -InAs from the origin of the two-valley structure in the $d^2I/dV^2 - V$ curves: *“It is obvious that the VB of p^+ -InAs cannot generate this large spin splitting ...correspond to the majority spin and minority spin CB of (In,Fe)As.”*
- ✧ In page 12, line 264 in the revised manuscript, we added a comment: *“This indicates that the Fe-related IB is irrelevant to the spin splitting observed in the tunnelling region of the two devices A and B.”*

2) As to alternative theories, Huang and Wessels noted that Fe in InAs is resonant with the conduction band see reference K. Huang *J. Appl. Phys* 64 6770 1988. From this they concluded that a vacuum referred binding energy (VRBE) model is relevant for transition metal doping of InAs and other III-V semiconductors. A model was subsequently proposed that transition metals with d-levels resonant with the semiconductor conduction or valence band should be a good FMS with high Curie temperatures (B. Wessels, *New Journal of Physics* 2008). Semiconductors with transition metals with d levels well within the band gap will not be good FMS as in the case of GaMnN. The InFeAs alloy studied here seems to support this VRBE model since the authors claim that Fe level is resonant with the conduction band.

The breakdown of the Zener model was discussed by Wessels, *New Journal of Physics* 2008. It is somewhat puzzling that the authors ignore the large body of literature on the InMnAs system in their introduction which has shown high T_c behavior. Also there is prior literature of Fe levels in InAs that should be discussed in light of their work.

The main conclusion is that there is major disagreement between the MF Zener theory of Dietl and Ohno and tunneling spectroscopy results presented in this work.

Our response:

We thank the reviewer for informing us of related literatures that we were unaware of, which helps us to improve the integrity and quality of our paper. We also agree with the suggestion of the reviewer and have revised the Discussion section to comply with it.

To estimate the energy position of the Fe levels in InAs, we have measured the band structure of (In,Fe)As (Fe 6%) using angle-resolved photoemission spectroscopy (ARPES), as shown in Fig. R11. By measuring at the resonant energy of Fe core-levels, we were able to observe the Fe-related IB close to the CB bottom of (In,Fe)As. This result is consistent with the observation of Huang and Wessels [*J. Appl. Phys* **64**, 6770 (1988)], despite the large difference in Fe concentration. Our TAMR results in the diffusion region (Fig. 4 in the main manuscript) also indicate that the Fe-related IB is close to the CB bottom or/and the VB top of (In,Fe)As, which is consistent with our ARPES results and those of Huang and Wessels.

We used the vacuum referred binding energy (VRBE) model to postulate the Fe-related IB levels in other semiconductors such as GaAs, GaSb, AlSb, Ge, using the known position of the Fe-related IB in InAs. Our experimental results show that, when

the Fe-related IB lies close to the CB bottom, as in (In,Fe)As, we observed strong n-type ferromagnetism. On the other hand, in (Ga,Fe)Sb [Refs. 34-36 in the main manuscript], (Al,Fe)Sb [L. D. Anh et al., *Appl. Phys. Lett.* **107**, 232405 (2015)], and GeFe [Wakabayashi, Y. K. et al. *Sci. Rep.* **6**, 23295 (2016)], where the Fe-related IB is close to the VB top, we observe strong p-type ferromagnetism. In contrast, when the Fe-related IB lies deep in the band gap as in the case of (Ga,Fe)As, the material is paramagnetic. Therefore, we came up with the model of “resonant *s,p-d* exchange interactions”, stating that the energy overlap of the *d*-band and the CB or VB of the host materials is important for realizing high- T_C FMSs, without learning the model previously proposed by B. W. Wessels [*New Journal of Physics* **10**, 055008 (17pp) (2008)]. Therefore, we agree with the reviewer that we should refer to those previous works.

Here we have significantly revised the Discussion section in the main manuscript. We introduce more experimental reports on high T_C narrow-gap FMSs such as (In,Mn)As, (In,Mn)Sb (grown by MOVPE) and (Ga,Fe)Sb (grown by MBE), where the mean-field Zener model failed to explain or predict the magnetic properties. We also cited some previous theoretical works that discussed the breakdown of the mean-field Zener model, and the resonant *s,p-d* model as an alternative approach.

Corresponding revised parts:

- ✧ In page 11, line 259 of the revised manuscript, we added a comment: “*This result is consistent with the observed position of the Fe deep levels in InAs, where Fe was doped at a very low concentration*³⁰.” We cited the work of K. Huang et al., *J. Appl. Phys* **64**, 6770 (1988) as ref. 30.
- ✧ We revised the Discussion part, from page 13, line 285: “*Besides the case of (In,Fe)As, the validity of the mean-field Zener model ...thus remains an unsolved theoretical challenge.*”, and cited works of (In,Mn)As, (In,Mn)Sb (grown by MOVPE) (ref. 37-39), and other theoretical papers on the break-down of the mean-field Zener model, including B. Wessels, *New Journal of Physics* **10**, 055008 (17pp) (2008) (ref. 31-33).

3) Other comments:

There is always confusion with possible magnetic precipitates in the Fe-As system. Are there any?

Note Be is an acceptor in III-V semiconductors see typo on line 73.

Our response:

We have conducted careful and systematic studies on the structural properties of

(In,Fe)As samples grown with the same conditions as the two samples A and B, using X-ray diffraction, high-resolution transmission electron microscopy (TEM), and three-dimensional atomic mapping, together with magnetization, magneto-optical, and magnetotransport characterizations. These results confirmed good zinc-blende crystal structure and absence of any second phase. The intrinsic origin of the ferromagnetism in (In,Fe)As thin films ($\text{Fe} \leq 9\%$) has also been inarguably confirmed. For references please take a look at our previous works (ref. 18, 19, 22, 23 in the revised manuscript).

On line 73 of the previous manuscript, we stated that Be dopants act as donors in (In,Fe)As grown by low-temperature MBE. This result has been confirmed in our previous work [see ref. 22, P. N. Hai *et al.*, *Appl. Phys. Lett.* **101**, 182403 (2012)]. Although Be is a well-known acceptor when replacing the group-III site in III-V semiconductors (including InAs), we have shown that when grown at low temperature ($\sim 250^\circ\text{C}$), Be dopants mainly become double-donors in (In,Fe)As layers, probably because they favorably sit in the interstitial sites.

Corresponding revised parts:

- ✧ In page 14, line 310 of the main manuscript, we have revised the explanation of the role of Be atoms in the Methods section as follows: " *In sample B we co-doped Be in the (In,Fe)As layer...because they favorably sit in the interstitial sites.*

Reviewers' Comments:

Reviewer #1 (Remarks to the Author):

I have read the revised manuscript and the author's replies. This paper represents high quality work. Along with all suggestions, they improved the manuscript. I recommend that this paper be accepted for publication in the present form.

Reviewer #2 (Remarks to the Author):

The authors have thoroughly responded to all reviewers comments with substantial revisions to the entire manuscript as well as new figures, modified figures, and supplementary information.

The manuscript is greatly improved and now suitable for publication in Nature Communications.

Reviewer #3 (Remarks to the Author):

The conduction properties of the ferromagnetic semiconductor InFeAs were investigated. Large spin splitting of the conduction band is observed in Esaki diodes comprised of n-type InFeAs and p-type InAs. Tunneling spectroscopy on two diodes indicated large spin splitting despite the predicted small value of s-d interaction. The measurements indicate that the mean-field Zener model underestimates the interaction by more than four orders of magnitude. This work supports prior findings that the Zener model does not adequately describe the interactions and properties of ferromagnetic semiconductors. Authors indicate that a comprehensive theory of ferromagnetic beyond the mean-field Zener model is needed.